# Multi-resolution Multi-task Gaussian Processes

**Oliver Hamelijnck**
The Alan Turing Institute
Department of Computer Science
University of Warwick
ohamelijnck@turing.ac.uk

**Theodoros Damoulas**
The Alan Turing Institute
Depts. of Computer Science & Statistics
University of Warwick
tdamoulas@turing.ac.uk

**Kangrui Wang**
The Alan Turing Institute
Department of Statistics
University of Warwick
kwang@turing.ac.uk

**Mark A. Girolami**
The Alan Turing Institute
Department of Engineering
University of Cambridge
mgirolami@turing.ac.uk

## Abstract

We consider evidence integration from potentially dependent observation processes under varying spatio-temporal sampling resolutions and noise levels. We offer a multi-resolution multi-task (MRGP) framework that allows for both *inter-task* and *intra-task* multi-resolution and multi-fidelity. We develop shallow Gaussian Process (GP) mixtures that approximate the difficult to estimate joint likelihood with a composite one and deep GP constructions that learn mappings between resolutions and naturally handle biases. In doing so, we generalize existing approaches and offer information-theoretic corrections and efficient variational approximations. We demonstrate the competitiveness of MRGPs on synthetic settings and on the challenging problem of hyper-local estimation of air pollution levels across London from multiple sensing modalities operating at disparate spatio-temporal resolutions.

## 1 Introduction

The increased availability of ground and remote sensing networks coupled with new modalities, arising from e.g. citizen science initiatives and mobile platforms, is creating new challenges for performing formal evidence integration. These multiple observation processes and sensing modalities can be dependent, with different signal-to-noise ratios and varying sampling resolutions across space and time. In our motivating application, London authorities measure air pollution from multiple sensor networks; high-fidelity ground sensors that provide frequent multi-pollutant readings, low fidelity diffusion tubes that only provide monthly single-pollutant readings, hourly satellite-derived information at large spatial scales, and high frequency medium-fidelity multi-pollutant sensor networks. Such a multi-sensor multi-resolution multi-task evidence integration setting is becoming prevalent across many real world applications of spatio-temporal problems.

The current state of the art, see also Section 5, is assuming independent and unbiased observation processes and cannot handle the challenges of real world settings that are jointly *non-stationary*, *multi-task*, *multi-fidelity*, and *multi-resolution* [2, 7, 14, 22, 23, 28, 29]. The latter challenge has recently attracted the interest of the machine learning community under the context of working with aggregate, binned observations [2, 14, 29] or the special case of natural language generation at multiple levels of abstraction [28]. When the independence and unbiasedness assumptions are not satisfied they lead to posterior contraction, degradation of predictive performance and insufficient uncertainty quantification.

In this paper we introduce a multi-resolution multi-task GP framework that can integrate evidence from observation processes with varying support (e.g. partially overlapping in time and space), that can be dependent and biased while allowing for both *inter-task* and *intra-task* multi-resolution and multi-fidelity. Our first contribution is a shallow GP mixture, MR-GPRN, that corrects for the dependency between observation processes through composite likelihoods and extends the Gaussian aggregation model of Law et al. [14], the multi-task GP model of Wilson et al. [33], and the variational lower bound of Nguyen and Bonilla [19]. Our second contribution is a multi-resolution deep GP composition that can additionally handle biases in the observation processes and extends the deep GP models and variational lower bounds of Damianou and Lawrence [5] and Salimbeni and Deisenroth [27] to varying support, multi-resolution data. Lastly, we demonstrate the superiority of our models on synthetic problems and on the challenging spatio-temporal setting of predicting air pollution in London at hyper-local resolution.

Sections 3 and 4 introduce our shallow GP mixtures and deep GP constructions, with associated variational approximations, respectively. In Section 6 we demonstrate the empirical advantages of our framework versus the prior art followed by additional related work in Section 5 and our concluding remarks. Further analysis is provided in the Appendix and code is available at `https://github.com/ohamelijnck/multi_res_gps`.

## 2 Multi-resolution Multi-task Learning

Consider $\mathcal{A} \in \mathbb{N}$ observation processes $\mathbf{Y}_a \in \mathbb{R}^{N_a \times P}$ across $P$ tasks with $N_a$ observations. Each process may be observed at varying resolutions that arises as the volume average over a sampling area $\mathcal{S}_a$. Typically we discretise the area $\mathcal{S}_a$ with a uniform grid and so we overload $\mathcal{S}_a$ to denote these points. We construct $\mathcal{A}$ datasets $\{(\mathbf{X}_a, \mathbf{Y}_a)\}_{a=1}^{\mathcal{A}}$, ordered by resolution size ($\mathbf{Y}_1$ is the highest, $\mathbf{Y}_{\mathcal{A}}$ is the lowest), where $\mathbf{X}_a \in \mathbb{R}^{N_a \times |\mathcal{S}_a| \times D_a}$ and $D_a$ is the input dimension. For notational simplicity we assume that all tasks are observed across all processes, although this need not be the case.

In our motivating application there are multiple sensor networks (observation processes) measuring multiple air pollutants (tasks) such as $CO_2$, $NO_2$, $PM_{10}$, $PM_{2.5}$ at different sampling resolutions. These multi-resolution observations exist both within tasks, (*intra-task multi-resolution*) when different sensor networks measure the same pollutant, and across tasks (*inter-task multi-resolution*) when different sensor networks measure different but potentially correlated pollutants due to e.g. common emission sources. Our goal is to develop scalable, non-stationary non-parametric models for air pollution while delivering accurate estimation and uncertainty quantification.

## 3 Multi-Resolution Gaussian Process Regression Networks (MR-GPRN)

We first introduce a *shallow* instantiation of the multi-resolution multi-task framework. MR-GPRN is a shallow GP mixture, Fig. 1, that extends the Gaussian process regression network (GPRN) [33]. Briefly, the GPRN jointly models all $P$ tasks as a linear combination of $Q \in \mathbb{N}$ GPs. These GPs are combined using task specific weights, that are themselves GPs, resulting in $PQ \in \mathbb{N}$ latent weights $\mathbf{W}_{\mathbf{p},\mathbf{q}}$. More formally, $\mathbf{f}_q \sim \mathcal{GP}(0, \mathbf{K}_q^f)$, $\mathbf{W}_{p,q} \sim \mathcal{GP}(0, \mathbf{K}_{p,q}^w)$ and each task $p$ is modelled as $\mathbf{Y}_p = \sum_{q=1}^{Q} \mathbf{W}_{p,q} \odot \mathbf{f}_q + \epsilon_p$ where $\odot$ is the Hadamard product and $\epsilon \sim \mathcal{N}(0, \sigma_p^2 \mathbf{I})$. The GPRN is an extension of the Linear Coregionalization Model (LCM) [3] and can enable the learning of non-stationary processes through input dependent weights [1].

### 3.1 Model Specification

We extend the GPRN model to handle multi-resolution observations by integrating the latent process over the sampling area for each observation. Apart from the standard inter-task dependency we would ideally want to be able to model additional dependencies between observation processes such as, for example, correlated noises. Directly modelling this additional dependency can quickly become intractable, due to the fact that it can vary in input space. If one ignores this dependency by assuming a product likelihood, as in [14, 18], then the misspecification results in severe posterior contractions (see Fig. 2). To circumvent these extremes we approximate the full likelihood using a multi-resolution composite likelihood that attempts to correct for this misspecification [31]. The posterior over the

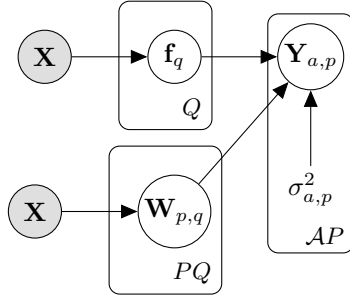

**Algorithm 1** Inference of MR-GPRN

**Input:** $\mathcal{A}$ multi-resolution datasets $\{(\mathbf{X}_a, \mathbf{Y}_a)\}_{a=1}^{\mathcal{A}}$, initial parameters $\theta$,
$\hat{\theta} \leftarrow \arg\max_\theta \sum_{a=1}^{\mathcal{A}} \ell(\mathbf{Y}_a|\theta)$
$\mathbf{H} \leftarrow \sum_{a=1}^{\mathcal{A}} \nabla\ell(\mathbf{Y}_a|\hat{\theta})\nabla\ell(\mathbf{Y}_a|\hat{\theta})^T$
$\mathbf{J} \leftarrow \nabla^2\ell(\mathbf{Y}|\hat{\theta})$
$\phi \leftarrow \begin{cases} \frac{|\hat{\theta}|}{\mathrm{Tr}[\mathbf{H}(\hat{\theta})^{-1}\mathbf{J}(\hat{\theta})]} \\ \frac{\mathrm{Tr}[\mathbf{H}(\hat{\theta})\mathbf{J}(\hat{\theta})^{-1}\mathbf{H}(\hat{\theta})]}{\mathrm{Tr}[\mathbf{H}(\hat{\theta})]} \end{cases}$
$\theta \leftarrow \arg\min_\theta \left( \sum_{a=1}^{\mathcal{A}} \phi \mathbb{E}_q\left[\ell(\mathbf{Y}_a|\theta)\right] + \mathcal{KL} \right)$

Figure 1: **Left**: Graphical model of MR-GPRN for $\mathcal{A}$ observation processes each with $P$ tasks. This allows *multi-resolution learning* between and across tasks. **Right**: Inference for MR-GPRN.

latent functions is now:

$$p(\mathbf{W}, \mathbf{f}|\mathbf{Y}) \propto \underbrace{\prod_{a=1}^{\mathcal{A}} \prod_{p=1}^{P} \prod_{n=1}^{N_a} \mathcal{N}\left(\mathbf{Y}_{a,p,n} \Big| \frac{1}{|\mathcal{S}_{a,n}|} \int_{\mathcal{S}_{a,n}} \sum_{q=1}^{Q} \mathbf{W}_{p,q}(\mathbf{x}) \odot \mathbf{f}_q(\mathbf{x}) \, d\mathbf{x}, \sigma_{a,p}^2 \mathbf{I}\right)^\phi}_{\text{MR-GPRN Composite Likelihood}} \underbrace{p(\mathbf{W}, \mathbf{f})}_{\text{GPRN Prior}} \quad (1)$$

where $\phi \in \mathbb{R}_{>0}$ are the composite weights that are important for inference. The integral within the multi-resolution likelihood links the underlying latent process to each of the resolutions; in general this is not available in closed form and so we approximate it by discretizing over a uniform grid. When we only have one task and $\mathbf{W}$ is set to a vector of constants we denote the model as MR-GP.

## 3.2 Composite Likelihood Weights

Under a misspecified model the asymptotic distribution of the MLE estimate converges to $\mathcal{N}(\theta_0, \frac{1}{n}\mathbf{H}(\theta_0)\mathbf{J}(\theta_0)^{-1}\mathbf{H}(\theta_0))$ where $\theta_0$ are the true parameters and $\mathbf{H}(\theta_0) = \frac{1}{n}\sum_{n=1}^{N} \nabla\ell(\mathbf{Y}|\theta_0)\nabla\ell(\mathbf{Y}|\theta_0)^T$, $\mathbf{J}(\theta_0) = \frac{1}{n}\sum_{n=1}^{N} \nabla^2\ell(\mathbf{Y}|\theta_0)$ are the Hessian and Jacobian respectively. The form of the asymptotic variance is the *sandwich information matrix* and it represents the loss of information in the MLE estimate due to the failure of Bartletts second identity [31].

Following Lyddon et al. [16] and Ribatet [26] we write down the asymptotic posterior of MR-GPRN as $\mathcal{N}(\theta_0, n^{-1}\phi^{-1}\mathbf{H}(\theta_0))$. In practise we only consider a subset of parameters that are present in all likelihood terms, such as the kernel parameters. Asymptotically one would expect the contribution of the prior to vanish causing the asymptotic posterior to match the limiting MLE. The composite weights $\phi$ can be used to bring these distributions as close together as possible. Approximating $\theta_0$ with the MLE estimate $\hat{\theta}$ and setting $\phi^{-1}\mathbf{H}(\hat{\theta}) = \mathbf{H}(\hat{\theta})\mathbf{J}(\hat{\theta})^{-1}\mathbf{H}(\hat{\theta})$ we can rearrange to find $\phi$ and recover the magnitude correction of Ribatet [26]. Instead if we take traces and then rearrange we recover the correction of Lyddon et al. [16]:

$$\phi_{\text{Ribatet}} = \frac{|\hat{\theta}|}{\mathrm{Tr}[\mathbf{H}(\hat{\theta})^{-1}\mathbf{J}(\hat{\theta})]} \quad , \quad \phi_{\text{Lyddon}} = \frac{\mathrm{Tr}[\mathbf{H}(\hat{\theta})\mathbf{J}(\hat{\theta})^{-1}\mathbf{H}(\hat{\theta})]}{\mathrm{Tr}[\mathbf{H}(\hat{\theta})]}. \quad (2)$$

## 3.3 Inference

In this section we derive a closed form variational lower bound for MR-GPRN, the full details can be found in the Appendix. For computational efficiency we introduce inducing points (see [10, 30]) $\mathbf{U} = \{\mathbf{u_q}\}_{q=1}^{Q}$ and $\mathbf{V} = \{\mathbf{v_{p,q}}\}_{p,q=1}^{P,Q}$, for the latent GPs $\mathbf{f}$ and $\mathbf{W}$ respectively, where $\mathbf{u_q} \in \mathbb{R}^M$ and $\mathbf{v_{p,q}} \in \mathbb{R}^M$. The inducing points are at the corresponding locations $\mathbf{Z}^{(\mathbf{u})} = \{\mathbf{Z_q^{(\mathbf{u})}}\}_{q=1}^{Q}, \mathbf{Z}^{(\mathbf{v})} = \{\mathbf{Z_{p,q}^{(\mathbf{v})}}\}_{p,q=1}^{P,Q}$ for $\mathbf{Z}_{\cdot}^{(\cdot)} \in \mathbb{R}^{M,D}$. We construct the augmented posterior and use the approximate

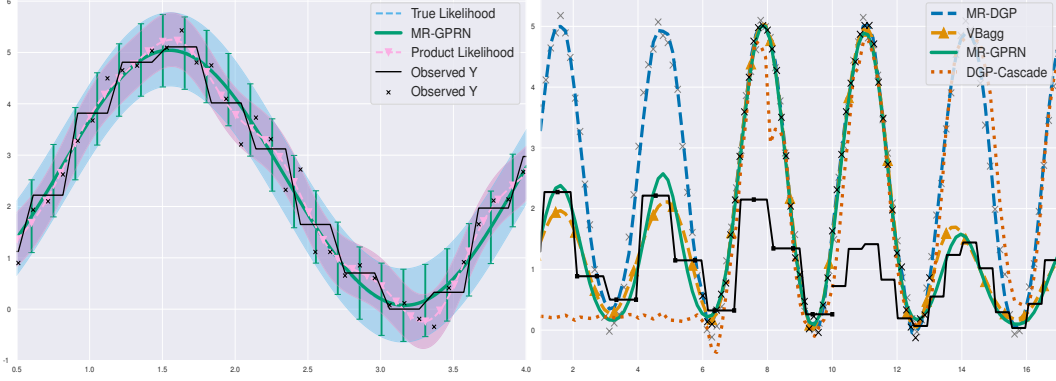

Figure 2: **Left**: MR-GPRN recovers the true predictive variance whereas assuming a product likelihood assumption leads to posterior contraction. **Right**: MR-DGP recovers the true predictive mean under a multi-resolution setting with scaling biases. Both VBAGG-NORMAL and MR-GPRN fail as they propagate the bias. Black crosses and lines denote observed values. Grey crosses denote observations removed for testing.

posterior $q(\mathbf{u}, \mathbf{v}, \mathbf{f}, \mathbf{W}) = p(\mathbf{f}, \mathbf{W}|\mathbf{u}, \mathbf{v})q(\mathbf{u}, \mathbf{v})$ where

$$q(\mathbf{u}, \mathbf{v}) = \sum_{k=1}^{K} \pi_k \prod_{j=1}^{Q} \mathcal{N}(\mathbf{m}_{k,j}^{(\mathbf{u})}, \mathbf{S}_{k,j}^{(\mathbf{u})}) \cdot \prod_{i,j=1}^{P,Q} \mathcal{N}(\mathbf{m}_{k,i,j}^{(\mathbf{v})}, \mathbf{S}_{k,i,j}^{(\mathbf{v})}) \tag{3}$$

is a free form mixture of Gaussians with $K$ components. We follow the variational derivation of [13, 21] and derive our expected log-likelihood $\mathrm{ELL} = \sum_{a=1}^{\mathcal{A}} \sum_{p=1}^{P} \sum_{n=1}^{N_a} \sum_{k=1}^{K} \mathrm{ELL}_{a,p,n,k}$,

$$\mathrm{ELL}_{a,p,n,k} = \pi_k \log \mathcal{N} \left( Y_{a,p,n} \mid \frac{1}{|\mathcal{S}_{a,n}|} \sum_{\mathbf{x} \in \mathcal{S}_{a,n}} \sum_{q=1}^{Q} \boldsymbol{\mu}_{k,p,q}^{(w)}(\mathbf{x}) \boldsymbol{\mu}_{k,q}^{(f)}(\mathbf{x}), \sigma_{a,p}^2 \right)$$
$$- \frac{\pi_k}{2\sigma_{a,p}^2} \frac{1}{|S_{a,n}|^2} \sum_{q=1}^{Q} \sum_{\mathbf{x}_1, \mathbf{x}_2} \boldsymbol{\Sigma}_{k,p,q}^{(w)} \boldsymbol{\Sigma}_{k,q}^{(f)} + \boldsymbol{\mu}_{k,q}^{(f)}(\mathbf{x}_1) \boldsymbol{\Sigma}_{k,p,q}^{(w)} \boldsymbol{\mu}_{k,q}^{(f)}(\mathbf{x}_2) \boldsymbol{\mu}_{k,p,q}^{(w)}(\mathbf{x}_1) \boldsymbol{\Sigma}_{k,q}^{(f)} \boldsymbol{\mu}_{k,p,q}^{(w)}(\mathbf{x}_2)$$
$$(4)$$

where $\boldsymbol{\Sigma}_{\cdot,\cdot,\cdot}^{(\cdot)}$ is evaluated at the points $\mathbf{x}_1, \mathbf{x}_2$. and $\boldsymbol{\mu}_k^{(f)}, \boldsymbol{\mu}_{k,p}^{(w)}, \boldsymbol{\Sigma}_k^{(f)}, \boldsymbol{\Sigma}_{k,p}^{(w)}$ are respectively the mean and variance of $q_k(\mathbf{W}_p), q_k(\mathbf{f})$. To infer the composite weights we follow [16, 26] and first obtain the MLE estimate of $\theta$ by maximizing the likelihood in Eq. 1. The weights can then be calculated and the variational lowerbound optimised as in Alg. 1 with $\mathcal{O}(E \cdot (PQ + Q)NM^2)$ for $E$ optimization steps until convergence. Our closed form ELBO generalizes prior state of the art of the GPRN ([1, 13, 19]) by extending to the multi-resolution setting and allowing for a free form mixture of Gaussians variational posterior. In the Appendix we also provide variational lower bounds for the positively-restricted GPRN form $\mathbf{Y}_p = \sum_{q=1}^{Q} \exp(\mathbf{W}_{p,q}) \odot \mathbf{f}_q + \epsilon$ that improves identifiability and predictive performance.

### 3.4 Prediction

Although the full predictive distribution of a specific observation process is not available in closed form, using the variational posterior we derive the predictive mean and variance, avoiding Monte Carlo estimates. The mean is simply $\mathbb{E}[\mathbf{Y}_{a,p}^*] = \sum_k^K \pi_k \mathbb{E}_k \left[ \mathbf{W}_{k,p}^* \right] \mathbb{E}_k[\hat{\mathbf{f}}_k^*]$, where $K$ is the number of components in the mixture of Gaussians variational posterior and $\pi_k$ is the $k$'th weight. We provide the predictive variance and full derivations in the appendix .

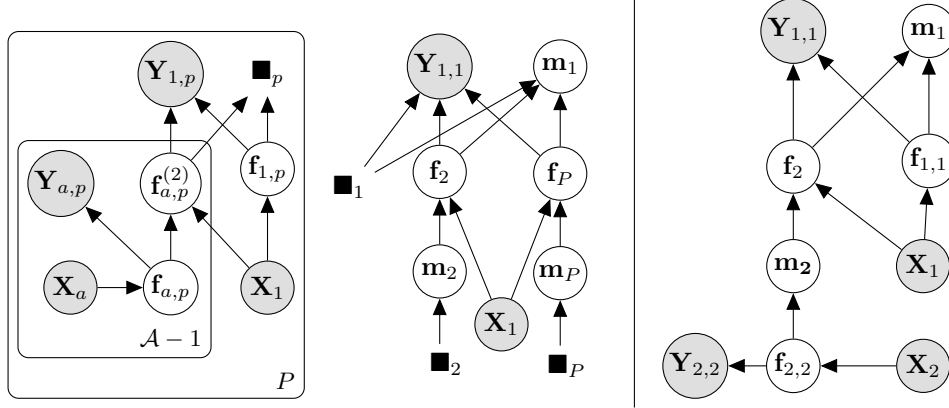

Figure 3: **Left**: General plate diagram of MR-DGP for $\mathcal{A}$ observation processes across $P$ tasks with noise variances omitted. For notational simplicity we have assumed that the target resolution is $a = 1$ and we use $\blacksquare_p$ to depict each of the sub-plate diagrams defined on the LHS. **Right**: A specific instantiation of an MR-DGP for 2 tasks and 2 observation processes (resolutions) with a target process $\mathbf{Y}_{1,1}$ as in the *inter*-task multi-resolution PM10, PM25 experiment in Section 4.

## 4    Multi-Resolution Deep Gaussian Processes (MR-DGP)

We now introduce MR-DGP, a deep instantiation of the framework which extends the deep GP (DGP) model of Damianou and Lawrence [5] into a tree-structured multi-resolution construction, Fig. 3. For notational convenience henceforth we assume that $p = 1$ is the target task and that $a = 1$ is the highest resolution and the one of primary interest. We note that this need not be the case and the relevant expressions can be trivially updated accordingly.

### 4.1    Model Specification

First we focus on the case when $P = 1$ and then generalize to an arbitrary number of tasks. We place $\mathcal{A}$ independent "Base" GPs $\{\mathbf{f}_{a,p}\}_{a=1}^{\mathcal{A}}$ on each of the $\mathcal{A}$ datasets within task $p$ that model their corresponding resolution independently. Taking $a = 1$ to be the target observation process we now construct $\mathcal{A} - 1$ two-layer DGPs that map from these base GPs $\{\mathbf{f}_{a,p}\}_{a=2}^{\mathcal{A}}$ to the target process $a = 1$ while learning an input-dependent mapping between observation processes. These DGPs are local experts that capture the information contained in each resolution for the target observation process. Every GP has an explicit likelihood which enables us to estimate and predict at every resolution and task while allowing for biases between observation processes to be corrected, see Fig. 2.

More formally, the joint distribution $p(\mathbf{Y}_p, \mathbf{F}_p)$ is given by:

$$\underbrace{\prod_{a=2}^{\mathcal{A}} \mathcal{N}(\mathbf{Y}_{1,p} | \frac{1}{|S_a|} \int_{S_a} \mathbf{f}_{a,p}^{(2)}(\mathbf{x}) \, d\mathbf{x}, \sigma_{a,p}^2) p(\mathbf{f}_{a,p}^{(2)} | \mathbf{f}_{a,p})}_{\text{Deep GPs}} \underbrace{\prod_{a=1}^{\mathcal{A}} \mathcal{N}((\mathbf{Y}_{a,p} | \frac{1}{|S_a|} \int_{S_a} \mathbf{f}_{a,p}(\mathbf{x}) \, d\mathbf{x}, \sigma_{a,p}^2) p(\mathbf{f}_{a,p})}_{\text{Base GPs}}$$

(5)

where $\mathbf{f}_{a,p} \sim \mathcal{GP}(0, \mathbf{K}_{a,p})$ and we have stacked all the observations and latent GPs into $\mathbf{Y}_p$ and $\mathbf{F}_p$ respectively. Each of the likelihood components is a special case of the multi-resolution likelihood in Eq. 1 (where $Q = 1$ and the latent GPs $\mathbf{W}$ are constant) and we discretize the integral in the same fashion. Similarly to the deep multi-fidelity model of [4] we define each DGP as:

$$p(\mathbf{f}_{a,p}^{(2)} | \mathbf{f}_{a,p}) = \mathcal{N}(0, \mathbf{K}_{a,p}^{(2)}((\mathbf{f}_{a,p}, \mathbf{X}_1), (\mathbf{f}_{a,p}, \mathbf{X}_1)))$$

(6)

where $\mathbf{X}_1$ are the covariates of the resolution of interest in our running example and allow each DGP to learn a mapping, between any observation process $a$ and the target one, that varies across $\mathbf{X}_1$. We now have $\mathcal{A}$ independent DGPs modelling $\mathbf{Y}_{1,p}$ with separable spatio-temporal kernels at each layer. The observation processes are not only at varying resolutions, but could also be partially overlapping or disjoint. This motivates treating each GP as a local model in a mixture of GP experts [35]. Mixture of GP experts typically combine the local GPs in two ways: either through a gating

network [24] or through weighing the local GPs [6, 20]. We employ the mixing weight approach in order to avoid the computational burden of learning the gating work. We define the mixture $\mathbf{m}_p = \beta_1 \odot \mathbf{f}_{1,p} + \sum_{a=1}^{\mathcal{A}} \beta_a \odot \mathbf{f}_{a,p}^{(2)}$ where the weight captures the reliability of the local GPs (or is set to 1 if the mixture is a singleton). The reliability is defined by the resolution and support of the base GPs and is naturally achieved by utilising the normalised log variances of the base GPs as $\beta_a = (1 - \mathbf{V}_a) \sum_i^a \mathbf{V}_i$. We provide the full justification and derivation for these weights in the appendix.

We can now generalize to an arbitrary number of tasks. For each task we construct a mixture of experts $\mathbf{m}_p$ as described above. For tasks $p > 1$ we learn the mapping from $\mathbf{m}_p$ to the target observation process $\mathbf{Y}_{1,1}$. This defines another set of local GP experts that is combined into a mixture with DGP experts. In our experiments we set $\mathbf{m}_p$ for $p > 1$ to be a simple average and for $\mathbf{m}_1$ we use our variance derived weights. This formulation naturally handles biases between the mean of different observations processes and each layer of the DGPs has a meaningful interpretation as it is modelling a specific observation process.

## 4.2 Augmented Posterior

Due to the non-linear forms of the parent GPs within the DGPs, marginalising out the parent GPs is generally analytically intractable. Following [27] we introduce inducing points $\mathbf{U} = \{\mathbf{u}_p\}_{p=2}^{P} \cup \{\mathbf{u}_{a,p}^{(2)}, \mathbf{u}_{a,p}\}_{a,p=1}^{P,\mathcal{A}}$ where each $\mathbf{u}_{\cdot,\cdot}^{(\cdot)} \in \mathbb{R}^M$ and inducing locations $\mathbf{Z} = \{\mathbf{Z}_p\}_{p=2}^{P} \cup \{\mathbf{Z}_{a,p}^{(2)}, \mathbf{Z}_{a,p}\}_{a,p=1}^{P,\mathcal{A}}$ where $\mathbf{Z}_p, \mathbf{Z}_{a,p}^{(2)} \in \mathbb{R}^{M \times (D+1)}$ and $\mathbf{Z}_{a,p} \in \mathbb{R}^{M \times D}$. The augmented posterior is now simply $p(\mathbf{Y}, \mathbf{F}, \mathbf{M}, \mathbf{U}) = p(\mathbf{Y}|\mathbf{F}) p(\mathbf{F}, \mathbf{M}|\mathbf{U}) p(\mathbf{U})$ where each $p(\mathbf{u}_{\cdot,\cdot}^{(\cdot)}) = \mathcal{N}(0, \mathbf{K}_{\cdot,\cdot}^{(\cdot)})$. Full details are provided in the appendix.

## 4.3 Inference

Following [27] we construct an approximate augmented posterior that maintains the dependency structure between layers:

$$q(\mathbf{M}, \mathbf{F}, \mathbf{U}) = p(\mathbf{M}, \mathbf{F}|\mathbf{U}) \prod_{p=2}^{P} q(\mathbf{u}_p) \cdot \prod_{p=1}^{P} \prod_{a=1}^{\mathcal{A}} q(\mathbf{u}_{a,p}^{(2)}) q(\mathbf{u}_{a,p}) \tag{7}$$

where each $q(\mathbf{u}_{\cdot,\cdot}^{(\cdot)})$ are independent free-form Gaussians $\mathcal{N}(\mathbf{m}_{\cdot,\cdot}^{(\cdot)}, \mathbf{S}_{\cdot,\cdot}^{(\cdot)})$ and the conditional is

$$p(\mathbf{F}, \mathbf{M}|\mathbf{U}) = \prod_{p=2}^{P} p(\mathbf{f}_p|\mathbf{m}_p, \mathbf{u}_p) p(\mathbf{m}_p|\mathrm{Pa}(\mathbf{m}_p)) \cdot \prod_{p=1}^{P} p(\mathbf{f}_{1,p}|\mathbf{u}_{1,p}) \prod_{a=2}^{\mathcal{A}} p(\mathbf{f}_{a,p}^{(2)}|\mathbf{f}_{a,p}, \mathbf{u}_{a,p}^{(2)}) p(\mathbf{f}_{a,p}|\mathbf{u}_{a,p}).$$
$$\tag{8}$$

We use $\mathrm{Pa}(\cdot)$ to denote the set of parent GPs of a given GP and $\mathcal{L}(\mathbf{f})$ to denote the depth of DGP $\mathbf{f}$, $p(\mathbf{m}_p|\mathrm{Pa}(\mathbf{m}_p)) = \mathcal{N}(\sum_a^{\mathcal{A}} \mathbf{w}_{a,p} \mu_{a,p}, \sum_a^{\mathcal{A}} \mathbf{w}_{a,p} \Sigma_{a,p} \mathbf{w}_{a,p})$ and $\mu_{a,p}, \Sigma_{a,p}$ are the means and variances of the relevant DGPs. Note that the mixture $\mathbf{m}_1$ combines all the DGPs at the top layer of the tree-hierarchy and hence it only appears in the predictive distribution of MR-DGP. All other terms are standard sparse GP conditionals and are provided in the Appendix. The ELBO is then simply derived as

$$\mathcal{L}_{\text{MR-DGP}} = \underbrace{\mathbb{E}_{q(\mathbf{M}, \mathbf{F}, \mathbf{U})} \left[ \log p(\mathbf{Y}|\mathbf{F}) \right]}_{\text{ELL}} + \underbrace{\mathbb{E}_{q(\mathbf{U})} \left[ \log \frac{P(\mathbf{U})}{q(\mathbf{U})} \right]}_{\text{KL}} \tag{9}$$

where the KL term is decomposed into a sum over all inducing variables $\mathbf{u}_{\cdot,\cdot}^{(\cdot)}$. The expected log likelihood (ELL) term decomposes across all $\mathbf{Y}$:

$$\sum_{p=2}^{P} \mathbb{E}_{q(\mathbf{f}_p)} \left[ \log p(\mathbf{Y}_{1,1}|\mathbf{f}_p) \right] + \sum_{p=1}^{P} \sum_{a}^{\mathcal{A}} \left[ \mathbb{E}_{q(\mathbf{f}_{a,1}^{(2)})} \left[ \log p(\mathbf{Y}_{1,p}|\mathbf{f}_{a,1}^{(2)}) \right] + \mathbb{E}_{q(\mathbf{f}_{a,p})} \left[ \log p(\mathbf{Y}_{a,p}|\mathbf{f}_{a,p}) \right] \right].$$
$$\tag{10}$$

For each ELL component the marginal $q(\mathbf{f}_{\cdot,\cdot}^{(\cdot)})$ is required. Because the base GPs are Gaussian, sampling is straightforward and the samples can be propagated through the layers, allowing the marginalization integral to be approximated by Monte Carlo samples. We use the reparametization trick to draw samples from the variational posteriors [11]. The inference procedure is given in Alg. 2.

---

**Algorithm 2** Inference procedure for MR-DGP

---

**Input:** $P$ multi-resolution datasets $\{(\mathbf{X}_p, \mathbf{Y}_p)\}_{p=1}^P$, initial parameters $\theta_0$,
**procedure** MARGINAL($\mathbf{f}$,$\mathbf{X}$, l, L)
    **if** $l = L$ **then**
        **return** $q(\mathbf{f}|\mathbf{X})$
    **end if**
    $q(\mathcal{P}(\mathbf{f})|\mathbf{X}) \leftarrow$ MARGINAL $(\mathcal{P}(\mathbf{f}), \mathbf{X}, l+1, \mathcal{L}(\mathcal{P}(\mathbf{f})))$
    **return** $\frac{1}{S}\sum_{s=1}^S p(\mathbf{f}|\mathbf{f}^{(s)}, \mathbf{X}))$ where $\mathbf{f}^{(s)} \sim q(\mathcal{P}(\mathbf{f})|\mathbf{X})$
**end procedure**
$\theta \leftarrow \arg\min_{\theta} \left[ \mathbb{E}_{\{\text{MARGINAL}(\mathbf{f}_p, \mathbf{X}_a, 0, \mathcal{L}(\mathbf{f}_p))\}_{p=1}^P} \left[\log p(\mathbf{Y}|\mathbf{F}, \mathbf{X}, \theta)\right] + \mathcal{KL}(q(\mathbf{U})||p(\mathbf{U})) \right]$

---

## 4.4 Prediction

**Predictive Density**. To predict at $\mathbf{x}^* \in \mathbb{R}^D$ in the target resolution $a = 1$ we simply approximate the predictive density $q(\mathbf{m}_1^*)$ by sampling from the variational posteriors and propagating the samples $\mathbf{f}^{(s)}$ through all the layers of the MR-DGP structure:

$$q(\mathbf{m}_1^*) = \int q(\mathbf{m}_1^*|\mathrm{Pa}(\mathbf{m}_1^*)) \prod_{\mathbf{f}\in\mathrm{Pa}(\mathbf{m}_1^*)} q(\mathbf{f})\, d\mathrm{Pa}(\mathbf{m}_1^*) \approx \frac{1}{S}\sum_{s=1}^S q(\mathbf{m}_1^*|\{\mathbf{f}^{(s)}\}_{\mathbf{f}\in\mathrm{Pa}(\mathbf{m}_1^*)}) \qquad (11)$$

In fact while propagating the samples through the tree structure the model naturally predicts at every resolution $a$ and task $p$ for the corresponding input location.

## 5 Related Work

Gaussian processes (GPs) are the workhorse for spatio-temporal modelling in spatial statistics [9] and in machine learning [25] with the direct link between multi-task GPs and Linear Models of Coregionalisation (LCM) reviewed by Alvarez et al. [3]. Heteroscedastic GPs [15] and recently proposed deeper compositions of GPs for the multi-fidelity setting [4, 22, 23] assume that all observations are of the same resolution. In spatial statistics the related *change of support* problem has been approached through Markov Chain Monte Carlo approximations and domain discretizations [8, 9]. Concurrently to our work [36] has explored the change of support problem under the setting of multi-variate areal data. They do not consider sparse GPs and hence can derive the true GP posterior, that is then approximated through an integral discretisation. A recent exception to discretising the multi-resolution integral is the work by Smith et al. [29] that solves the integral for squared exponential kernels but only considers observations from one resolution and cannot handle additional input features. Independently, and concurrently, [34] have recently proposed a multi-resolution LCM model that is similar to our MR-GPRN model without dependent observation processes and composite likelihood corrections. Instead they focus on improved estimation of the area integral and non-Gaussian likelihoods. Finally, we note that the multi-resolution GP work by Fox and Dunson [7] defines a DGP construction for non-stationary models that is more akin to multi-scale modelling [32]. This line of research typically focuses on learning multiple kernel lengthscales to explain both broad and fine variations in the underlying process and hence cannot handle multi-resolution observations .

## 6 Experiments

We demonstrate and evaluate the MRGPs on synthetic experiments and the challenging problem of estimating and forecasting air pollution in the city of London. We compare against VBAGG-NORMAL [14] and two additional baselines. The first, CENTER-POINT , is a GPRN modified to support multi-resolution data by representing each aggregation region through its centre point only. The second, MR-CASCADE is an instance of MR-DGP but, to illustrate the benefits of the tree composition and the mixture of experts approach of MR-DGP, instead of a tree structured DGP (as in Fig. 3) we

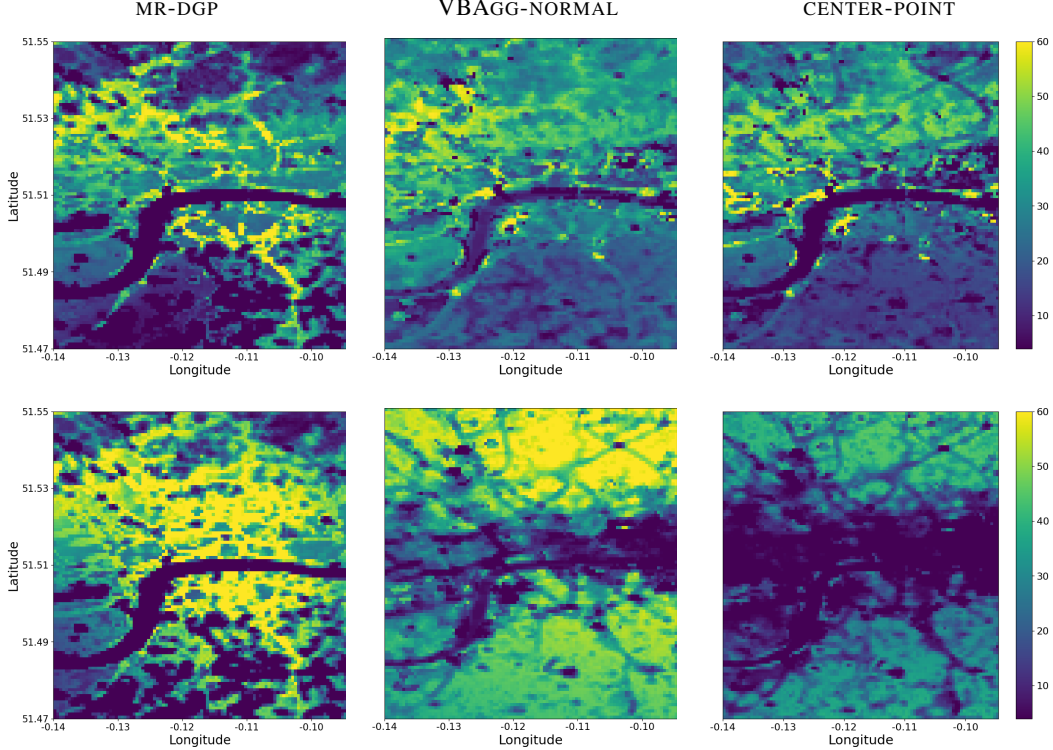

Figure 4: Spatio-temporal estimation and forecasting of $NO_2$ levels in London. **Top Row**: Spatial slices from MR-GPRN, VBAGG-NORMAL and CENTER-POINT respectively at 19/02/2019 11:00:00 using observations from both LAQN and the satellite model (low spatial resolution). **Bottom Row**: Spatial slices at the base resolution from the same models at 19/02/2019 17:00:00 where *only* observations from the satellite model are present.

construct a cascade. Experiments are coded[1] in *TensorFlow* and we provide additional analysis in the Appendix.

**Dependent observation processes**: We provide details of the dependent observation processes experiment in the left of Fig. 2 in the Appendix.

**Biased observation processes:**. To demonstrate the ability of MR-DGP in handling biases across observation processes we construct 3 datasets from the function $\mathbf{y} = s \cdot 5 \sin(\mathbf{x})^2 + 0.1\epsilon$ where $\epsilon \sim \mathcal{N}(0, 1)$. The first $\mathbf{X}_1, \mathbf{Y}_1$ is at resolution $\mathcal{S}_1 = 1$ in the range x=[7,12] with a scale $s = 1$. The second is at resolution of $\mathcal{S}_2 = 5$ between x=[-10, 10] with a scale $s = 0.5$ and lastly the third is at resolution of $\mathcal{S}_3 = 5$ x=[10, 20] with a scale $s = 0.3$. The aim is to predict $\mathbf{y}$ across the range [-10, 20] and the results are shown in Table 2 and Fig. 2. MR-DGP significantly outperforms all of the four alternative approaches as it is learning a forward *mapping* between observations.

**Training**. When training both MR-GPRN and VBAGG-NORMAL we first jointly optimize the variational and hyper parameters while keeping the likelihood variances fixed and then jointly optimize all parameters together. For MR-DGP we first optimize layer by layer and then jointly optimize all parameters together, see appendix, as we find that this helps to avoid early local optima.

***Inter*-task multi-resolution: modelling of $PM_{10}$ and $PM_{25}$ in London**: In this experiment we consider multiple tasks with different resolutions. We jointly model $PM_{10}$ and $PM_{25}$ at a specific LAQN location in London. The site we consider is *RB7* in the date range 18/06/2018 to 28/06/2018. At this location we have hourly data from both $PM_{10}$ and $PM_{25}$. To simulate having multiple resolutions we construct 2, 5, 10 and 24 hour aggregations of $PM_{10}$ and remove a 2 day region of

Table 1: *Inter*-task multi-resolution. Missing data predictive MSE on PM$_{25}$ from MR-GPRN, MR-DGP and baseline CENTER-POINT for 4 different aggregation levels of PM$_{10}$. VBAGG-NORMAL is inapplicable in this experiment as it is a single-task approach.

| Model | PM$_{10}$ Resolution | | | |
|---|---|---|---|---|
| | 2 Hours | 5 Hours | 10 Hours | 24 Hours |
| CENTER-POINT | $4.67 \pm 0.74$ | $5.04 \pm 0.45$ | $5.26 \pm 0.91$ | $5.72 \pm 0.91$ |
| MR-GPRN | $4.54 \pm 0.93$ | $5.09 \pm 1.04$ | $4.96 \pm 1.07$ | $5.32 \pm 1.14$ |
| MR-DGP | $5.14 \pm 1.28$ | $4.81 \pm 1.06$ | $4.61 \pm 1.43$ | $5.42 \pm 1.15$ |

Table 2: *Intra*-task multi-resolution. **Left**: Predicting NO$_2$ across London (Fig. 4). **Right**: Synthetic experiment results (Fig. 2) with three observations processes and scaling bias.

| Model | RMSE | MAPE | Model | RMSE | MAPE |
|---|---|---|---|---|---|
| Single GP | $20.55 \pm 9.44$ | $0.8 \pm 0.16$ | MR-CASCADE | 2.12 | 0.16 |
| CENTER-POINT | $18.74 \pm 12.65$ | $0.65 \pm 0.21$ | VBAGG-NORMAL | 1.68 | 0.14 |
| VBAGG-NORMAL | $16.16 \pm 9.44$ | $0.69 \pm 0.37$ | MR-GPRN | 1.6 | 0.14 |
| MR-GPRN w/o CL | $12.97 \pm 9.22$ | $0.56 \pm 0.32$ | MR-DGP | **0.19** | **0.02** |
| MR-GPRN w CL | $11.92 \pm 6.8$ | $0.45 \pm 0.17$ | | | |
| MR-DGP | $\mathbf{6.27 \pm 2.77}$ | $\mathbf{0.38 \pm 0.32}$ | | | |

PM$_{25}$ which is the test region. The results from all of our models in Table 1 demonstrate the ability to successfully learn the multi-task dependencies. Note that CENTER-POINT fails, e.g. Table 2, when the sampling area cannot be approximated by a single center point due to the scale of the underlying process.

***Intra*-task multi-resolution: spatio-temporal modelling of NO$_2$ in London**: In this experiment we consider the case of a single task but with multiple multi-resolution observation processes. First we use observations coming from ground point sensors from the London Air Quality Network (LAQN). These sensors provide hourly readings of NO$_2$. Secondly we use observations arising from a global satellite model [17] that provide hourly data at a spatial resolution of 7km × 7km and provide 48 hour forecasts. We train on both the LAQN and satellite observations from 19/02/2018-20/02/2018 and the satellite ones from 20/02/2018-21/02/2018. We then predict at the resolution of the LAQN sensors in the latter date range. To calculate errors we predict for each LAQN sensor site, and find the average and standard deviation across all sites.

We find that MR-DGP is able to substantially outperform both VBAGG-NORMAL, MR-GPRN and the baselines, Table 2 (left), as it is learning the forward mapping between the low resolution satellite observations and the high resolution LAQN sensors, while handling scaling biases. This is further highlighted in the bottom of Fig. 4 where MR-DGP is able to retain high resolution structure based only on satellite observations whereas VBAGG-NORMAL and CENTER-POINT over-smooth.

# 7 Conclusion

We offer a framework for evidence integration when observation processes can have varying *inter*- and *intra-task* sampling resolutions, dependencies, and different signal to noise ratios. Our motivation comes from a challenging and impactful problem of hyper-local air quality prediction in the city of London, while the underlying multi-resolution multi-sensor problem is general and pervasive across modern spatio-temporal settings and applications of machine learning. We proposed both shallow mixtures and deep learning models that generalise and outperform the prior art, correct for posterior contraction, and can handle biases in observation processes such as discrepancies in the mean. Further directions now open up to robustify the multi-resolution framework against outliers and against further model misspecification by exploiting ongoing advances in generalized variational inference [12]. Finally an open challenge remains on developing continuous model constructions that avoid domain discretization, as in [2, 34], for more complex settings.

**Acknowledgements**

O. H., T. D and K.W. are funded by the Lloyd's Register Foundation programme on Data Centric Engineering through the London Air Quality project. This work is supported by The Alan Turing Institute for Data Science and AI under EPSRC grant EP/N510129/1 in collaboration with the Greater London Authority. We would like to thank the anonymous reviewers for their feedback and Libby Rogers, Patrick O'Hara, Daniel Tait and Juan Maroñas for their help on multiple aspects of this work.

## Footnotes

[1]Codebase and datasets to reproduce results are available at `https://github.com/ohamelijnck/multi_res_gps`

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
