[Supplementary Material]

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

# A  MR-GPRN

In this section we provide the full derivation of the variational lower bound for MR-GPRN. Recall that we have a multi-resolution data set across $P$ tasks and we model each task as

$$\mathbf{Y}_{a,p} = \frac{1}{|\mathcal{S}_a|} \int_{\mathcal{S}_a} \sum_{q=1}^{Q} \mathbf{W}_{p,q}(\mathbf{x})\mathbf{f}_q(\mathbf{x})\,d\mathbf{x} + \epsilon_{a,p} \tag{12}$$

where $\epsilon_{a,p} \sim \mathcal{N}(0, \sigma_{a,p}^2 \mathbf{I})$. The joint density $p(\mathbf{Y}, \mathbf{W}, \mathbf{f})$ is then proportional to

$$\underbrace{\prod_{a=1}^{\mathcal{A}}\prod_{p=1}^{P}\prod_{n=1}^{N_a}\mathcal{N}(\mathbf{Y}_{a,p,n}|\frac{1}{|\mathcal{S}_{a,n}|}\int_{\mathcal{S}_{a,n}}\sum_{q=1}^{Q}\mathbf{W}_{p,q}(\mathbf{x})\odot\mathbf{f}_q(\mathbf{x})\,d\mathbf{x}, \sigma_a^2\mathbf{I})^\phi}_{\text{MR-GPRN Composite Likelihood}}\underbrace{\prod_{p=1}^{P}\prod_{q=1}^{Q}p(\mathbf{W}_{a,p})\prod_{q=1}^{Q}p(\mathbf{f}_q)}_{\text{GPRN Prior}}. \tag{13}$$

where $\mathbf{f}_q \sim \mathcal{N}(0, \mathbf{K}_q)$ are global functions across all tasks and $\mathbf{W}_{p,q} \sim \mathcal{N}(0, \mathbf{K}_{p,q})$ are task specific. To allow for computationally efficient inference we introduce inducing points for all latent functions [32]. For the latent basis functions, $\mathbf{f}$, we have the inducing points $\mathbf{u} = \{\mathbf{u}_q\}_{q=1}^{Q}$ where $\mathbf{u}_q \in \mathbb{R}^{M_q}$ at locations $\mathbf{Z}^{(f)} = \{\mathbf{Z}_q\}_{q=1}^{Q}$ for $\mathbf{Z}_q \in \mathbb{R}^{M_q,D}$. Similarly, for the latent weight functions, $\mathbf{W}$, we have $\mathbf{v} = \{\mathbf{v}_{p,q}\}_{p,q=1}^{P,Q}$ where $\mathbf{v}_{p,q} \in \mathbb{R}^{M_{p,q}}$ at locations $\mathbf{Z}^{(w)} = \{\mathbf{Z}_{p,q}\}_{p,q=1}^{P,Q}$ for $\mathbf{Z}_{p,q} \in \mathbb{R}^{M_{p,q},D}$. We assume that $\mathbf{f}$ and $\mathbf{W}$ are independent Gaussian processes, and furthermore that they factor across components. We then then write the joint probability density including the inducing points as

$$\begin{aligned}
p(\mathbf{Y}, \mathbf{W}, \mathbf{f}, \mathbf{u}, \mathbf{v}) &\propto p(\mathbf{Y}|\mathbf{W}, \mathbf{f})p(\mathbf{W}, \mathbf{f}|\mathbf{u}, \mathbf{v})p(\mathbf{u}, \mathbf{v}) \\
&= p(\mathbf{Y}|\mathbf{W}, \mathbf{f})p(\mathbf{f}|\mathbf{u})p(\mathbf{W}|\mathbf{v})p(\mathbf{u})p(\mathbf{v}) \\
&= p(\mathbf{Y}|\mathbf{W}, \mathbf{f})\prod_{p=1}^{P}\prod_{q=1}^{Q}p(\mathbf{W}_{p,q}|\mathbf{v}_{p,q})p(\mathbf{v}_{p,q})\prod_{q=1}^{Q}p(\mathbf{f}_q|\mathbf{u}_q)p(\mathbf{u}_q)
\end{aligned} \tag{14}$$

where $p(\mathbf{f}_q|\mathbf{u}_q) = \mathcal{N}(\mathbf{f}_q|\boldsymbol{\mu}_q'^{(u)}, \boldsymbol{\Sigma}_q'^{(u)})$ and $p(\mathbf{W}_{p,q}|\mathbf{v}_{p,q}) = \mathcal{N}(\mathbf{W}_{p,q}|\boldsymbol{\mu}_{p,q}'^{(v)}, \boldsymbol{\Sigma}_{p,q}'^{(v)})$ have the standard conditional Gaussian distributions with parameters

$$\begin{aligned}
\boldsymbol{\mu}_q'^{(u)} &= \mathbf{K}_q^{(u)}(\mathbf{X}, \mathbf{Z}_q)\mathbf{K}_q^{(u)}(\mathbf{Z}_q, \mathbf{Z}_q)^{-1}\mathbf{u}_q \\
\boldsymbol{\mu}_{p,q}'^{(v)} &= \mathbf{K}_{p,q}^{(v)}(\mathbf{X}, \mathbf{Z}_{p,q})\mathbf{K}_{p,q}^{(v)}(\mathbf{Z}_{p,q}, \mathbf{Z}_{p,q})^{-1}\mathbf{v}_{p,q} \\
\boldsymbol{\Sigma}_{\cdot,\cdot}'^{(\cdot)} &= \mathbf{K}_{\cdot,\cdot}^{(\cdot)}(\mathbf{X}, \mathbf{X}) - \mathbf{K}_{\cdot,\cdot}^{(\cdot)}(\mathbf{X}, \mathbf{Z}_{\cdot,\cdot})\mathbf{K}_{\cdot,\cdot}^{(\cdot)}(\mathbf{Z}_{\cdot,\cdot}, \mathbf{Z}_{\cdot,\cdot})^{-1}\mathbf{K}_{\cdot,\cdot}^{(\cdot)}(\mathbf{Z}_{\cdot,\cdot}, \mathbf{X})
\end{aligned} \tag{15}$$

and $p(\mathbf{u}_q) = \mathcal{N}(\mathbf{u}_q|0, \mathbf{K}_q^{(u)})$ and $p(\mathbf{v}_{p,q}) = \mathcal{N}(\mathbf{v}_{p,q}|0, \mathbf{K}_{p,q}^{(v)})$.

## A.1  Approximate Posterior

Variational inference turns the Bayesian posterior inference problem into an optimisation problem where the objective function is the evidence lower bound (ELBO). Following [10] we construct our variational lower bound, keeping $\mathbf{u}$ and $\mathbf{v}$ explicit, allowing for stochastic variational inference. We define our approximate posterior to have the factorisation

$$q(\mathbf{u}, \mathbf{v}, \mathbf{f}, \mathbf{W}) = p(\mathbf{f}, \mathbf{W}|\mathbf{u}, \mathbf{v})q(\mathbf{u}, \mathbf{v}) \tag{16}$$

where we have defined $q(\mathbf{u}, \mathbf{v})$ to be a free-form mixture of Gaussians

$$q(\mathbf{u}, \mathbf{v}) = \sum_{k=1}^{K} \pi_k \prod_{j=1}^{Q} \mathcal{N}(\mathbf{u}_j|\mathbf{m}_{k,j}^{(\mathbf{u})}, \mathbf{S}_{k,j}^{(\mathbf{u})}) \cdot \prod_{i,j=1}^{P,Q} \mathcal{N}(\mathbf{v}_{i,j}|\mathbf{m}_{k,i,j}^{(\mathbf{v})}, \mathbf{S}_{k,i,j}^{(\mathbf{v})}) \tag{17}$$

where each mean $\mathbf{m}_k \in \mathbb{R}^M$ and covariance factor $\mathbf{S}_k \in \mathbb{R}^{M \times M}$. The marginal $q(\mathbf{f}, \mathbf{W})$ is then derived as

$$
\begin{aligned}
q(\mathbf{f}, \mathbf{W}) &= \int p(\mathbf{f}, \mathbf{W}|\mathbf{u}, \mathbf{v})q(\mathbf{u}, \mathbf{v})\,d\mathbf{u}\,d\mathbf{v} \\
&= \sum_{k=1}^{K} \pi_k \int \prod_{q=1}^{Q} p(\mathbf{f}_q|\mathbf{u}_{k,q})q(\mathbf{u}_{k,q})\,d\mathbf{u}_{k,q} \cdot \int \prod_{p,q=1}^{P,Q} p(\mathbf{W}_{p,q}|\mathbf{v}_{k,p,q})q(\mathbf{v}_{k,p,q})\,d\mathbf{v}_{k,p,q} \\
&= \sum_{k=1}^{K} \pi_k \prod_{q=1}^{Q} \mathcal{N}(\mathbf{f}_q|\boldsymbol{\mu}_{k,q}^{(f)}, \boldsymbol{\Sigma}_{k,q}^{(f)}) \cdot \prod_{p,q=1}^{P,Q} \mathcal{N}(\mathbf{W}_{p,q}|\boldsymbol{\mu}_{k,p,q}^{(W)}, \boldsymbol{\Sigma}_{k,p,q}^{(W)})
\end{aligned}
\tag{18}
$$

where the moments of the variational distribution are given by

$$
\begin{aligned}
\boldsymbol{\mu}_{k,\cdot,\cdot}^{(\cdot)} &= \mathbf{K}^{(\cdot)}(\mathbf{X}, \mathbf{Z}_\cdot)\mathbf{K}^{(\cdot)}(\mathbf{Z}_\cdot, \mathbf{Z}_\cdot)^{-1}\mathbf{m}_{k,\cdot}^{(\cdot)} \\
\boldsymbol{\Sigma}_{k,\cdot,\cdot}^{(\cdot)} &= \boldsymbol{\Sigma}_{k,\cdot,\cdot}^{\prime(\cdot)} + \mathbf{K}^{(\cdot)}(\mathbf{X}, \mathbf{Z}_\cdot)\mathbf{K}^{(\cdot)}(\mathbf{Z}_\cdot, \mathbf{Z}_\cdot)^{-1}\mathbf{S}_{k,\cdot,\cdot}^{(\cdot)}\mathbf{K}^{(\cdot)}(\mathbf{Z}_\cdot, \mathbf{Z}_\cdot)^{-1}\mathbf{K}^{(\cdot)}(\mathbf{Z}_\cdot, \mathbf{X})
\end{aligned}
\tag{19}
$$

## A.2 Variational Lower Bound

Following [14] we derive the ELBO as:

$$
\begin{aligned}
\mathcal{L}_{\text{MR-GPRN}} &= \mathbb{E}_q\left[\log \frac{p(\mathbf{Y}, \mathbf{W}, \mathbf{f}, \mathbf{u}, \mathbf{v})}{q(\mathbf{W}, \mathbf{f}, \mathbf{u}, \mathbf{v}))}\right] \\
&= \mathbb{E}_q\left[\log \frac{p(\mathbf{Y}|\mathbf{W}, \mathbf{f})p(\mathbf{f}|\mathbf{u})p(\mathbf{u})p(\mathbf{W}|\mathbf{v})p(\mathbf{v})}{p(\mathbf{f}|\mathbf{u})p(\mathbf{W}|\mathbf{v})q(\mathbf{u}, \mathbf{v}))}\right] \\
&= \underbrace{\mathbb{E}_{q(\mathbf{f}, \mathbf{W})}\left[\log p(\mathbf{Y}|\mathbf{f}, \mathbf{W})\right]}_{\text{ELL}} + \underbrace{\mathbb{E}_{q(\mathbf{u}, \mathbf{v})}\left[\log \frac{p(\mathbf{u}, \mathbf{v})}{q(\mathbf{u}, \mathbf{v})}\right]}_{KL}
\end{aligned}
\tag{20}
$$

The subsequent sections derive the closed-form expressions of both the expected log likelihood (ELL) and the KL term.

## A.3 MR-GPRN: KL Term

Following [14] the KL term is decomposed into two terms:

$$
\text{KL} = \mathbb{E}_q\left[\log \frac{p(\mathbf{u}, \mathbf{v})}{q(\mathbf{u}, \mathbf{v})}\right] = \underbrace{\mathbb{E}_q\left[\log p(\mathbf{u}, \mathbf{v})\right]}_{\text{cross}} - \underbrace{\mathbb{E}_q\left[\log q(\mathbf{u}, \mathbf{v})\right]}_{\text{ent}}
\tag{21}
$$

where we deal with each term separately.

### A.3.1 Cross Term

The cross term is calculated as

$$
\begin{aligned}
\text{cross} &= \mathbb{E}_q\left[\log p(\mathbf{u}, \mathbf{v})\right] = \sum_{k=1}^{K} \pi_k \mathbb{E}_{q_k(\mathbf{u}, \mathbf{v})}\left[\log p(\mathbf{u}, \mathbf{v})\right] \\
&= \sum_{k=1}^{K} \pi_k \left(\sum_{q=1}^{Q} \mathbb{E}_{q(\mathbf{u}_{k,q})}\left[\log p(\mathbf{u}_q)\right] + \sum_{p=1}^{P}\sum_{q=1}^{Q} \mathbb{E}_{q(\mathbf{v}_{k,p,q})}\left[\log p(\mathbf{v}_{p,q})\right]\right)
\end{aligned}
\tag{22}
$$

where each

$$
\begin{aligned}
\mathbb{E}_{q(\mathbf{u}_{k,q})}\left[\log p(\mathbf{u}_q)\right] &= C_u - \frac{1}{2}(\mathbf{m}_{k,q}^{(m)})^T\mathbf{K}_q^u(\mathbf{Z}_q^{(u)}, \mathbf{Z}_q^{(u)})^{-1}\mathbf{m}_{k,q}^{(u)} - \frac{1}{2}Tr\left[\mathbf{K}_q^u(\mathbf{Z}_q^{(u)}, \mathbf{Z}_q^{(u)})^{-1}\mathbf{S}_{k,q}^u\right] \\
\mathbb{E}_{q(\mathbf{v}_{k,p,q})}\left[\log p(\mathbf{v}_{p,q})\right] &= C_v - \frac{1}{2}(\mathbf{m}_{k,p,q}^{(v)})^T\mathbf{K}_{p,q}^v(\mathbf{Z}_{p,q}^{(v)}, \mathbf{Z}_{p,q}^{(v)})^{-1}\mathbf{m}_{k,p,q}^{(v)} - \frac{1}{2}Tr\left[\mathbf{K}_{p,q}^v(\mathbf{Z}_{p,q}^{(v)}, \mathbf{Z}_{p,q}^{(v)})^{-1}\mathbf{S}_{k,p,q}^v\right]
\end{aligned}
\tag{23}
$$

and

$$C_u = -\frac{M_q}{2}\log(2\pi) - \frac{1}{2}\log|\mathbf{K}_q^{(u)}(\mathbf{Z}_{\mathbf{q}}^{(\mathbf{u})}, \mathbf{Z}_{\mathbf{q}}^{(\mathbf{u})})|$$
$$C_v = -\frac{M_{p,q}}{2}\log(2\pi) - \frac{1}{2}\log|\mathbf{K}_{p,q}^{(v)}(\mathbf{Z}_{\mathbf{p,q}}^{(\mathbf{v})}, \mathbf{Z}_{\mathbf{p,q}}^{(\mathbf{v})})| \tag{24}$$

### A.3.2 Entropy Term

Following [14] we lower bound the entropy term of the mixture of Gaussians as

$$
\begin{aligned}
\text{ent} &= -\sum_{k=1}^{K}\pi_k\mathbb{E}_{q_k(\mathbf{u},\mathbf{v})}\left[\log q(\mathbf{u},\mathbf{v})\right] \\
&\geq -\sum_{k=1}^{K}\pi_k\log\left(\mathbb{E}_{q_k(\mathbf{u},\mathbf{v})}\left[q(\mathbf{u},\mathbf{v})\right]\right) \\
&= -\sum_{k=1}^{K}\pi_k\log\left(\sum_{l=1}^{K}\pi_l\mathbb{E}_{q_k(\mathbf{u})}[q_l(\mathbf{u})]\cdot\mathbb{E}_{q_k(\mathbf{v})}[q_l(\mathbf{v})]\right) \\
&= -\sum_{k=1}^{K}\pi_k\log\left(\sum_{l=1}^{K}\pi_l\mathcal{N}(\mathbf{m}_k^{(u)}|\mathbf{m}_l^{(u)}, \mathbf{S}_k^{(u)} + \mathbf{S}_l^{(u)})\cdot\mathcal{N}(\mathbf{m}_k^{(v)}|\mathbf{m}_l^{(v)}, \mathbf{S}_k^{(v)} + \mathbf{S}_l^{(v)})\right).
\end{aligned}
\tag{25}
$$

### A.4 MR-GPRN: Closed Form Expected Log Likelihood

We now derive the closed form expected log likelihood (ELL) in Eq. 20. The ELL is

$$\text{ELL} = \sum_{a=1}^{\mathcal{A}}\sum_{p=1}^{P}\sum_{n=1}^{N_a}\sum_{k=1}^{K}\pi_k\underbrace{\mathbb{E}_{q_k(\mathbf{f},\mathbf{W})}\left[\log\mathcal{N}(\mathbf{Y}_{a,p,n}|\frac{1}{|\mathcal{S}_{a,n}|}\sum_{\mathbf{x}\in\mathcal{S}_{a,n}}\sum_{q=1}^{Q}\mathbf{W}_{p,q}(\mathbf{x})\mathbf{f}_q(\mathbf{x}), \sigma_a^2)\right]}_{\text{ELL}_a} \tag{26}$$

where each of the components can now be dealt with separately. Dealing with component $a$

$$
\begin{aligned}
\text{ELL}_a &= \mathbb{E}_{q_k(\mathbf{f},\mathbf{W})}\left[\log\mathcal{N}(\mathbf{Y}_{a,p,n}|\frac{1}{|\mathcal{S}_{a,n}|}\sum_{\mathbf{x}\in\mathcal{S}_{a,n}}\sum_{q=1}^{Q}\mathbf{W}_{p,q}(\mathbf{x})\mathbf{f}_q(\mathbf{x}), \sigma_a^2)\right] \\
&= C_1 + C_2\mathbb{E}_{q_k(\mathbf{f},\mathbf{W})}\left[(\mathbf{Y}_{a,p,n} - \mu_y)^T(\mathbf{Y}_{a,p,n} - \mu_y)\right] \\
&= C_1 + C_2\left(\mathbb{E}_{q_k(\mathbf{f},\mathbf{W})}\left[\mathbf{Y}_{a,p,n}^T\mathbf{Y}_{a,p,n}\right] - 2\cdot\mathbb{E}_{q_k(\mathbf{f},\mathbf{W})}\left[\mathbf{Y}_{a,p,n}\mu_y\right] + \mathbb{E}_{q_k(\mathbf{f},\mathbf{W})}\left[\mu_y^2\right]\right)
\end{aligned}
\tag{27}
$$

where

$$
\begin{aligned}
\mu_y &= \frac{1}{|\mathcal{S}_{a,n}|}\sum_{\mathbf{x}\in\mathcal{S}_{a,n}}\sum_{q=1}^{Q}\mathbf{W}_{p,q}(\mathbf{x})\mathbf{f}_q(\mathbf{x}) \\
C_1 &= -\frac{N_a}{2}\log(2\pi\sigma_{a,p}^2) \\
C_2 &= -\frac{1}{2\sigma_{a,p}^2}
\end{aligned}
\tag{28}
$$

and we now deal with each of these expectations separately.

### A.4.1 ELL: 1st Term

The first expectation does not contain $\mathbf{f}$ or $\mathbf{W}$ and so the expectations can be dropped

$$\mathbb{E}_{q_k(\mathbf{f},\mathbf{W})}\left[\mathbf{Y}_{a,p,n}^2\right] = \mathbf{Y}_{a,p,n}^2 \tag{29}$$

### A.4.2 ELL: 2nd Term

For the second term the expectation is brought inside the sum, and then applied to each of $\mathbf{f}$ and $\mathbf{W}$ separately

$$
\begin{aligned}
\mathbb{E}_{q_k(\mathbf{f},\mathbf{W})}\left[\mu_y \mathbf{Y}_{a,p,n}\right] &= \left(\frac{1}{|\mathcal{S}_{a,n}|}\sum_{\mathbf{x}\in\mathcal{S}_{a,n}}\sum_{q=1}^{Q}\mathbb{E}_q\left[\mathbf{W}_{p,q}(\mathbf{x})\right]\mathbb{E}_q\left[\mathbf{f}_q(\mathbf{x})\right]\right)\mathbf{Y}_{a,p,n}\\
&= \left(\frac{1}{|\mathcal{S}_{a,n}|}\sum_{\mathbf{x}\in\mathcal{S}_{a,n}}\sum_{q=1}^{Q}\boldsymbol{\mu}_{k,p,q}^{(w)}(\mathbf{x})\boldsymbol{\mu}_{k,q}^{(f)}(\mathbf{x})\right)\mathbf{Y}_{a,p,n}
\end{aligned}
\tag{30}
$$

where we have used the independence property, $q(\mathbf{f},\mathbf{W}) = q(\mathbf{f})q(\mathbf{W})$.

### A.4.3 ELL: 3rd Term

In the last expectation we have a product of sums, that is then expanded into a quadruple sum over $q_1$, $q_2$. There will be two cases; the first is when $q_1 = q_2$ inside the sum $\mathbf{f}$ and $\mathbf{W}$ will appear in square forms, and the second when the expectations can be treated as in Sec. A.4.2.

The final term obtained after expanding the quadratic in (27) is given by

$$
\mathbb{E}_{q_k(\mathbf{f},\mathbf{W})}\left[\mu_y^2\right] = \frac{1}{|\mathcal{S}_{a,n}|^2}\sum_{q_1=1}^{Q}\sum_{q_2=1}^{Q}\sum_{\mathbf{x}_1}^{\mathcal{S}_{a,n}}\sum_{\mathbf{x}_2}^{\mathcal{S}_{a,n}}\mathbb{E}_{q_k(\mathbf{f},\mathbf{W})}\left[\mathbf{f}_{q_1}(\mathbf{x}_1)\mathbf{W}_{p,q_1}(\mathbf{x}_1)\mathbf{W}_{p,q_2}(\mathbf{x}_2)\mathbf{f}_{q_2}(\mathbf{x}_2)\right]
\tag{31}
$$

For the case where $q_1 = q_2$, then upon taking expectations we have

$$
\begin{aligned}
&\mathbb{E}_{q_k(\mathbf{f},\mathbf{W})}\left[(\mathbf{f}_{q_1}(\mathbf{x}_1)\mathbf{W}_{p,q_1}(\mathbf{x}_1)\mathbf{W}_{p,q_1}(\mathbf{x}_2)\mathbf{f}_{q_1}(\mathbf{x}_2)\right]\\
&= \mathbb{E}_{q_k(\mathbf{f})}\left[\mathbf{f}_{q_1}(\mathbf{x}_1)\left(\boldsymbol{\mu}_{k,p,q_1}^{(w)}(\mathbf{x}_1)\boldsymbol{\mu}_{k,p,q_1}^{(w)}(\mathbf{x}_2) + \boldsymbol{\Sigma}_{k,p,q_1}^{(w)}(\mathbf{x}_1,\mathbf{x}_2)\right)\mathbf{f}_{q_1}(\mathbf{x}_2)\right]\\
&= \boldsymbol{\Sigma}_{k,p,q_1}^{(w)}(\mathbf{x}_1,\mathbf{x}_2)\boldsymbol{\Sigma}_{k,q_1}^{(f)}(\mathbf{x}_1,\mathbf{x}_2) + \boldsymbol{\mu}_{k,q_1}^{(f)}(\mathbf{x}_1)\boldsymbol{\Sigma}_{k,p,q_1}^{(w)}(\mathbf{x}_1,\mathbf{x}_2)\boldsymbol{\mu}_{k,q_1}^{(f)}(\mathbf{x}_2)+\\
&\quad \boldsymbol{\mu}_{k,p,q_1}^{(w)}(\mathbf{x}_1)\boldsymbol{\Sigma}_{k,q_1}^{(f)}(\mathbf{x}_1,\mathbf{x}_2)\boldsymbol{\mu}_{k,p,q_1}^{(w)}(\mathbf{x}_2) + \boldsymbol{\mu}_{k,q_1}^{(f)}(\mathbf{x}_1)\boldsymbol{\mu}_{k,p,q_1}^{(w)}(\mathbf{x}_1)\boldsymbol{\mu}_{k,p,q_1}^{(w)}(\mathbf{x}_2)\boldsymbol{\mu}_{k,q_1}^{(f)}(\mathbf{x}_2)
\end{aligned}
\tag{32}
$$

where we use the notation $\boldsymbol{\Sigma}_{\cdot,\cdot,\cdot}^{\cdot}(\mathbf{x}_1,\mathbf{x}_2)$ to denote entry of the covariance matrix which agrees with the enumeration of $\mathbf{x}_1$ and $\mathbf{x}_2$. And similarly $\boldsymbol{\mu}_{k,p,q_1}^{(w)}(\mathbf{x}_1)$, $\boldsymbol{\mu}_{k,\cdot}^{(\cdot)}(\mathbf{x}_1)$ denotes the entry of the mean vector agreeing with the enumeration of $\mathbf{x}_1$.

When $q_1 \neq q_2$ there will be no square terms and so the expectation will simply be

$$
\mathbb{E}_{q_k(\mathbf{f},\mathbf{W})}\left[\mathbf{f}_{q1}(\mathbf{x}_1)\mathbf{W}_{p,q1}(\mathbf{x}_1)\mathbf{W}_{p,q2}(\mathbf{x}_2)\mathbf{f}_{q2}(\mathbf{x}_2)\right] = \boldsymbol{\mu}_{k,q_1}^{(f)}(\mathbf{x}_1)\boldsymbol{\mu}_{k,p,q_1}^{(w)}(\mathbf{x}_1)\boldsymbol{\mu}_{k,p,q_1}^{(w)}(\mathbf{x}_2)\boldsymbol{\mu}_{k,q_1}^{(f)}(\mathbf{x}_2)
\tag{33}
$$

The complete third term can now be rewritten as

$$
\begin{aligned}
\mathbb{E}_q\left[\mu_y^2\right] =& \frac{1}{|\mathcal{S}_{a,n}|^2}\sum_{q_1=1}^{Q}\sum_{q_2=1}^{Q}\sum_{\mathbf{x}_1}^{\mathcal{S}_{a,n}}\sum_{\mathbf{x}_2}^{\mathcal{S}_{a,n}}\boldsymbol{\mu}_{k,q_1}^{(f)}(\mathbf{x}_1)\boldsymbol{\mu}_{k,p,q_1}^{(w)}(\mathbf{x}_1)\boldsymbol{\mu}_{k,p,q_1}^{(w)}(\mathbf{x}_2)\boldsymbol{\mu}_{k,q_1}^{(f)}(\mathbf{x}_2)+\\
& \frac{1}{|\mathcal{S}_{a,n}|^2}\sum_{q=1}^{Q}\sum_{\mathbf{x}_1}^{\mathcal{S}_{a,n}}\sum_{\mathbf{x}_2}^{\mathcal{S}_{a,n}}\boldsymbol{\Sigma}_{k,p,q}^{(w)}(\mathbf{x}_1,\mathbf{x}_2)\boldsymbol{\Sigma}_{k,q}^{(f)}(\mathbf{x}_1,\mathbf{x}_2)\\
& + \boldsymbol{\mu}_{k,q}^{(f)}(\mathbf{x_1})\boldsymbol{\Sigma}_{k,p,q}^{(w)}(\mathbf{x}_1,\mathbf{x}_2)\boldsymbol{\mu}_{k,q}^{(f)}(\mathbf{x}_2) + \boldsymbol{\mu}_{k,p,q}^{(w)}(\mathbf{x}_1)\boldsymbol{\Sigma}_{k,q}^{(f)}(\mathbf{x}_1,\mathbf{x}_2)\boldsymbol{\mu}_{k,p,q}^{(w)}(\mathbf{x}_2)
\end{aligned}
\tag{34}
$$

### A.4.4 Full Term

Combining the derivations above we have

$$
\begin{aligned}
\text{ELL} = \sum_{a=1}^{\mathcal{A}} \sum_{k=1}^{K} \pi_k \frac{1}{2\sigma_a^2} &\Bigg[ \mathbf{Y}_{a,p,n}^2 - 2 \left( \frac{1}{|\mathcal{S}_{a,n}|} \sum_{\mathbf{x}\in\mathcal{S}_{a,n}} \sum_{q=1}^{Q} \boldsymbol{\mu}_{k,p,q}^{(w)}(\mathbf{x}) \boldsymbol{\mu}_{k,q}^{(f)}(\mathbf{x}) \right) \mathbf{Y}_{a,p,n} \\
&+ \frac{1}{|\mathcal{S}_{a,n}|^2} \Bigg( \sum_{q_1,q_2=1}^{Q} \sum_{\mathbf{x}_1,\mathbf{x}_2}^{\mathcal{S}_{a,n}} \boldsymbol{\mu}_{k,q_1}^{(f)}(\mathbf{x}_1) \boldsymbol{\mu}_{k,p,q_1}^{(w)}(\mathbf{x}_1) \boldsymbol{\mu}_{k,p,q_1}^{(w)}(\mathbf{x}_2) \boldsymbol{\mu}_{k,q_1}^{(f)}(\mathbf{x}_2) \\
&+ \delta_{q_1,q_2} \cdot \Big( \boldsymbol{\Sigma}_{k,p,q}^{(w)}(\mathbf{x}_1,\mathbf{x}_2) \boldsymbol{\Sigma}_{k,q}^{(f)}(\mathbf{x}_1,\mathbf{x}_2) + \boldsymbol{\mu}_{k,q}^{(f)}(\mathbf{x}_1) \boldsymbol{\Sigma}_{k,p,q}^{(w)}(\mathbf{x}_1,\mathbf{x}_2) \boldsymbol{\mu}_{k,q}^{(f)}(\mathbf{x}_2) \boldsymbol{\mu}_{k,p,q}^{(w)}(\mathbf{x}_1) \boldsymbol{\Sigma}_{k,q}^{(f)}(\mathbf{x}_1,\mathbf{x}_2) \boldsymbol{\mu}_{k,p,q}^{(w)}(\mathbf{x}_2) \Big) \Bigg) \Bigg] \\
&- \frac{1}{2} \sum_{a=1}^{\mathcal{A}} N_a P \log(2\pi\sigma_a^2).
\end{aligned}
$$

or

$$
\begin{aligned}
\text{ELL} = \sum_{a=1}^{A} \sum_{p=1}^{P} \sum_{n=1}^{N_a} \sum_{k=1}^{K} &\pi_k \log \mathcal{N} \left( Y_{a,p,n} \mid \frac{1}{|\mathcal{S}_{a,n}|} \sum_{\mathbf{x}\in\mathcal{S}_{a,n}} \sum_{q=1}^{Q} \boldsymbol{\mu}_{k,p,q}^{(w)}(\mathbf{x}) \boldsymbol{\mu}_{k,q}^{(f)}(\mathbf{x}), \sigma_{a,p}^2 \right) \\
&- \sum_{a=1}^{A} \sum_{p=1}^{P} \sum_{n=1}^{N_a} \sum_{k=1}^{K} \frac{\pi_k}{2\sigma_{a,p}^2} \frac{1}{|S_{a,n}|^2} \sum_{q=1}^{Q} \sum_{\mathbf{x}_1,\mathbf{x}_2} \boldsymbol{\Sigma}_{k,p,q}^{(w)}(\mathbf{x}_1,\mathbf{x}_2) \boldsymbol{\Sigma}_{k,q}^{(f)}(\mathbf{x}_1,\mathbf{x}_2) \\
&+ \boldsymbol{\mu}_{k,q}^{(f)}(\mathbf{x}_1) \boldsymbol{\Sigma}_{k,p,q}^{(w)}(\mathbf{x}_1,\mathbf{x}_2) \boldsymbol{\mu}_{k,q}^{(f)}(\mathbf{x}_2) \boldsymbol{\mu}_{k,p,q}^{(w)}(\mathbf{x}_1) \boldsymbol{\Sigma}_{k,q}^{(f)}(\mathbf{x}_1,\mathbf{x}_2) \boldsymbol{\mu}_{k,p,q}^{(w)}(\mathbf{x}_2)
\end{aligned}
\tag{35}
$$

### A.5 MR-GPRN: Closed Form Expected Log Likelihood $(\mathbf{W} \to \exp(\mathbf{W}))$

#### A.5.1 Moments of exponentiated Gaussian random variables

In this section we provide additional results needed for the calculations in this section using standard properties of the moment generating function of a Gaussian random variable.

$$
\begin{aligned}
\mathbb{E}[\exp(t \cdot \mathbf{W}_{p,q})] &= \int_{\mathbb{R}} \mathcal{N}(\mathbf{W}_{p,q} \mid \mu_{\mathbf{W}_{p,q}}, \Sigma_{\mathbf{W}_{p,q}}) \exp(t \cdot \mathbf{W}_{p,q}) \, d\mathbf{W}_{p,q} \\
&= \exp\left(t\mu_{\mathbf{W}_{p,q}} + \frac{t^2}{2}\Sigma_{\mathbf{W}_{p,q}}\right)
\end{aligned}
\tag{36}
$$

where $\exp(\cdot)$ is defined as element-wise function, for arbitrary $t \in \mathbb{R}$.

#### A.5.2 ELL with positive weights

If $\mathbf{W}$ is passed through an exponential function to enforce positive latent weights the expected log-likelihood is

$$
\alpha \sum_{a=1}^{A} \sum_{p=1}^{P} \sum_{n=1}^{N_a} \sum_{k=1}^{K} \pi_k \, \underbrace{\mathbb{E}_{q_k(\mathbf{f},\mathbf{W})} \left[ \log \mathcal{N}(\mathbf{Y}_{a,p,n} \mid \frac{1}{|\mathcal{S}_{a,n}|} \sum_{\mathbf{x}\in\mathcal{S}_{a,n}} \sum_{q=1}^{Q} \exp(\mathbf{W}_{p,q}(\mathbf{x})) \mathbf{f}_q(\mathbf{x}), \sigma_a^2) \right]}_{\text{ELL}_a}
\tag{37}
$$

Each of the likelihood components can now be dealt with separately, and moreover has the same form as Eq. 27 with

$$
\mu_y = \frac{1}{|\mathcal{S}_{a,n}|} \sum_{\mathbf{x}\in\mathcal{S}_{a,n}} \sum_{q=1}^{Q} \exp(\mathbf{W}_{p,q}(\mathbf{x})) \mathbf{f}_q(\mathbf{x})
\tag{38}
$$

As above, we now deal with the expectation of each term obtained after expanding the quadratic separately, but using the results for the exponentiated moments in A.5.1.

### A.5.3 ELL: 1st Term

As in Sec. A.4.1 the expectation is constant:

$$\mathbb{E}_{q_k(\mathbf{f},\mathbf{W})}\left[\mathbf{Y}_{a,p,n}^2\right] = \mathbf{Y}_{a,p,n}^2 \tag{39}$$

### A.5.4 ELL: 2nd Term

As in Sec. A.4.2 the 2nd term the expectation is brought inside the sum and applied to $\mathbf{f}$ and $\mathbf{W}$ separately. To evaluate the expectation of $\exp(\mathbf{W})$ we use the result from Eq. 36:

$$
\begin{aligned}
\mathbb{E}_{q_k(\mathbf{f},\mathbf{W})}\left[\mu_y \mathbf{Y}_{a,p,n}\right] &= \left(\frac{1}{|\mathcal{S}_{a,n}|}\sum_{\mathbf{x}\in\mathcal{S}_{a,n}}\sum_{q=1}^{Q}\mathbb{E}_{q_k(\mathbf{f},\mathbf{W})}\left[\exp(\mathbf{W}_{p,q}(\mathbf{x}))\right]\mathbb{E}_{q_k(\mathbf{f},\mathbf{W})}\left[\mathbf{f}_q(\mathbf{x})\right]\right)\mathbf{Y}_{a,p,n} \\
&= \left(\frac{1}{|\mathcal{S}_{a,n}|}\sum_{\mathbf{x}\in\mathcal{S}_{a,n}}\sum_{q=1}^{Q}\exp\left(\boldsymbol{\mu}_{k,p,q}^{(w)}(\mathbf{x})+\frac{1}{2}\boldsymbol{\Sigma}_{k,p,q}^{(w)}(\mathbf{x},\mathbf{x})\right)\boldsymbol{\mu}_{k,q}^{(f)}(\mathbf{x})\right)\mathbf{Y}_{a,p,n}.
\end{aligned}
\tag{40}
$$

### A.5.5 ELL: 3rd Term

This derivation closely follows that from Section A.4.3. The third term is given by

$$\mathbb{E}_{q_k(\mathbf{f},\mathbf{W})}\left[\mu_y^2\right] = \frac{1}{|\mathcal{S}_{a,n}|^2}\sum_{q_1=1}^{Q}\sum_{q_2=1}^{Q}\sum_{\mathbf{x}_1}^{\mathcal{S}_{a,n}}\sum_{\mathbf{x}_2}^{\mathcal{S}_{a,n}}\mathbb{E}_{q_k(\mathbf{f},\mathbf{W})}\left[\mathbf{f}_{q_1}(\mathbf{x}_1)\exp\left(\mathbf{W}_{p,q_1}(\mathbf{x}_1)\right)\exp\left(\mathbf{W}_{p,q_2}(\mathbf{x}_2)\right)\mathbf{f}_{q_2}(\mathbf{x}_2)\right] \tag{41}$$

For the case where $q_1 = q_2$

$$
\begin{aligned}
\mathbb{E}_{q_k(\mathbf{f},\mathbf{W})}&\left[\mathbf{f}_{q_1}(\mathbf{x}_1)\exp\left(\mathbf{W}_{p,q_1}(\mathbf{x}_1)\right)\exp\left(\mathbf{W}_{p,q_1}(\mathbf{x}_2)\right)\mathbf{f}_{q_1}(\mathbf{x}_2)\right] \\
&=\mathbb{E}_{q_k(\mathbf{f})}\left[\mathbf{f}_{q_1}(\mathbf{x}_1)\left(\widetilde{\boldsymbol{\mu}}_{k,p,q_1}^{(w)}(\mathbf{x}_1)\widetilde{\boldsymbol{\mu}}_{k,p,q_1}^{(w)}(\mathbf{x}_2)+\widetilde{\boldsymbol{\Sigma}}_{k,p,q}^{(w)}(\mathbf{x}_1,\mathbf{x}_2)\right)\mathbf{f}_{q_1}(\mathbf{x}_2)\right] \\
&=\widetilde{\boldsymbol{\Sigma}}_{k,p,q_1}^{(w)}(\mathbf{x}_1,\mathbf{x}_2)\boldsymbol{\Sigma}_{k,q_1}^{(f)}(\mathbf{x}_1,\mathbf{x}_2)+\boldsymbol{\mu}_{k,q_1}^{(f)}(\mathbf{x}_1)\widetilde{\boldsymbol{\Sigma}}_{k,p,q_1}^{(w)}(\mathbf{x}_1,\mathbf{x}_2)\boldsymbol{\mu}_{k,q_1}^{(f)}(\mathbf{x}_2)+ \\
&\quad\widetilde{\boldsymbol{\mu}}_{k,p,q_1}^{(w)}(\mathbf{x}_1)\boldsymbol{\Sigma}_{k,q_1}^{(f)}(\mathbf{x}_1,\mathbf{x}_2)\widetilde{\boldsymbol{\mu}}_{k,p,q_1}^{(w)}(\mathbf{x}_2)+\boldsymbol{\mu}_{k,q_1}^{(f)}(\mathbf{x}_1)\widetilde{\boldsymbol{\mu}}_{k,p,q_1}^{(w)}(\mathbf{x}_1)\widetilde{\boldsymbol{\mu}}_{k,p,q_1}^{(w)}(\mathbf{x}_2)\boldsymbol{\mu}_{k,q_1}^{(f)}(\mathbf{x}_2)
\end{aligned}
\tag{42}
$$

where

$$\widetilde{\boldsymbol{\mu}}_{k,p,q}^{(w)}(\mathbf{x}_1) = \exp(\boldsymbol{\mu}_{k,p,q}^{(w)}(\mathbf{x}_1)+\frac{1}{2}\boldsymbol{\Sigma}_{k,p,q}^{(w)}(\mathbf{x}_1,\mathbf{x}_1)),$$

and

$$\widetilde{\boldsymbol{\Sigma}}_{k,p,q}^{(w)}(\mathbf{x}_1,\mathbf{x}_2) = \exp(2(\boldsymbol{\mu}_{k,p,q}^{(w)}+\boldsymbol{\Sigma}_{k,p,q}^{(w)}(\mathbf{x}_1,\mathbf{x}_2))).$$

In all other cases there will be no square terms and so the expectation will simply be

$$
\begin{aligned}
\mathbb{E}_{q_k(\mathbf{f},\mathbf{W})}&[\mathbf{f}_{q1}(\mathbf{x}_1)\exp(\mathbf{W}_{p,q1}(\mathbf{x}_1))\exp(\mathbf{W}_{p,q2}(\mathbf{x}_2))\mathbf{f}_{q2}(\mathbf{x}_2)] \\
&= \boldsymbol{\mu}_{k,q1}^{(f)}(\mathbf{x}_1)\widetilde{\boldsymbol{\mu}}_{k,p,q1}^{(w)}(\mathbf{x}_1)\widetilde{\boldsymbol{\mu}}_{k,p,q2}^{(w)}(\mathbf{x}_2)\boldsymbol{\mu}_{k,q2}^{(f)}(\mathbf{x}_2)
\end{aligned}
\tag{43}
$$

### A.6 Prediction

Although the full predictive distribution for MR-GPRN is not available analytically, we are able to derive the first and second moments in closed form.

### A.6.1 Predictive Mean

To calculate the first moment we use the approximate variational posterior in place of the true posterior. The predictive mean for task $p$ is given as

$$
\begin{aligned}
\mathbb{E}[\mathbf{y}_p^* \mid \mathbf{x}^*, \mathbf{X}, \mathbf{Y}] &= \int \mathbf{y}_p^* \, p(\mathbf{y}_p^* \mid \mathbf{X}^*, \mathbf{X}, \mathbf{Y}) \, d\mathbf{y}^* \\
&\approx \sum_{k=1}^{K} \pi_k \int \mathbf{y}_p^* \, p(\mathbf{y}_p^* \mid \mathbf{W}_{k,p}^*, \mathbf{f}_k^*, \mathbf{x}^*, \mathbf{X}, \mathbf{Y}) q(\mathbf{f}_k^*) q(\mathbf{W}_k^*) \, d\mathbf{y}_p^* \, d\mathbf{W}_{k,p}^* \, d\mathbf{f}_k^* \\
&= \sum_{k=1}^{K} \pi_k \int \sum_{q=1}^{Q} \mathbf{W}_{p,q}^* \mathbf{f}_q^* q(\mathbf{f}_k^*) q(\mathbf{W}_k^*) \, d\mathbf{W}_{k,p}^* \, d\mathbf{f}_k^* \\
&= \sum_{k=1}^{K} \pi_k \sum_{q=1}^{Q} \boldsymbol{\mu}_{k,p,q}^{(w)}(\mathbf{x}^*) \boldsymbol{\mu}_{k,q}^{(f)}(\mathbf{x}^*)
\end{aligned}
\tag{44}
$$

### A.6.2 Predictive Variance

The point wise second moment of task $p$ is given by

$$
\mathbb{V}[\mathbf{y}_p^*] = \mathbb{E}\left[(\mathbf{y}_p^*)^2\right] - \mathbb{E}\left[\mathbf{y}_p^*\right] \mathbb{E}\left[\mathbf{y}_p^*\right]
\tag{45}
$$

We have already calculated the closed form mean in Eq. 44 and the square form is given by

$$
\begin{aligned}
\mathbb{E}\left[(\mathbf{y}_p^*)^2\right] &\approx \sum_{k=1}^{K} \pi_k \int (\mathbf{y}_p^*)^2 p(\mathbf{y}_p^* | \mathbf{W}_{k,p}^*, \mathbf{f}_k^*, \mathbf{x}^*, \mathbf{X}, \mathbf{Y}) q(\mathbf{f}_k^*) q(\mathbf{W}_k^*) \, d\mathbf{y}_p^* \, d\mathbf{W}_{k,p}^* \, d\mathbf{f}_k^* \\
&= \sum_{k=1}^{K} \pi_k \sigma_{\mathbf{y}_p}^2 I + \sum_{k=1}^{K} \pi_k \mathbb{E}_{q_k(\mathbf{f}^*, \mathbf{W}^*)} \left[\left(\sum_{q_1=1}^{Q} \mathbf{W}_{k,p,q_1}^* \mathbf{f}_{k,q_1}^*\right) \left(\sum_{q_2=1}^{Q} \mathbf{W}_{k,p,q_2}^* \mathbf{f}_{k,q_2}^*\right)\right]
\end{aligned}
\tag{46}
$$

In the case when $q_1 = q_2$ the expectation is

$$
\begin{aligned}
\sum_{q_1=1}^{Q} &\mathbb{E}_{q_k(\mathbf{f}^*, \mathbf{W}^*)} \left[\mathbf{W}_{k,p,q}^* \mathbf{f}_{k,q}^* \mathbf{f}_{k,q}^* \mathbf{W}_{k,p,q}^*\right] \\
&= \sum_{q_1=1}^{Q} \mathbb{E}_{q_k(\mathbf{W}^*)} \left[\mathbf{W}_{k,p,q_1}^* \left(\boldsymbol{\mu}_{k,q_1}^{(f)}(\mathbf{x}^*) \boldsymbol{\mu}_{k,q_1}^{(f)}(\mathbf{x}^*) + \boldsymbol{\Sigma}_{k,q_1}^{(f)}(\mathbf{x}^*, \mathbf{x}^*)\right) \mathbf{W}_{k,p,q_1}^*\right] \\
&= \sum_{q_1=1}^{Q} \left[\boldsymbol{\mu}_{k,p,q_1}^{(W)}(\mathbf{x}^*) \boldsymbol{\mu}_{k,q_1}^{(f)}(\mathbf{x}^*) \boldsymbol{\mu}_{k,q_1}^{(f)}(\mathbf{x}^*) \boldsymbol{\mu}_{k,p,q_1}^{(W)}(\mathbf{x}^*) + \boldsymbol{\mu}_{k,p,q_1}^{(W)}(\mathbf{x}^*) \boldsymbol{\Sigma}_{k,q_1}^{(f)}(\mathbf{x}^*, \mathbf{x}^*) \boldsymbol{\mu}_{k,p,q_1}^{(W)}(\mathbf{x}^*)\right. \\
&\left. + \boldsymbol{\mu}_{k,q_1}^{(f)}(\mathbf{x}^*) \boldsymbol{\Sigma}_{k,p,q_1}^{(W)}(\mathbf{x}^*, \mathbf{x}^*) \boldsymbol{\mu}_{k,q_1}^{(f)}(\mathbf{x}^*) + \boldsymbol{\Sigma}_{k,q_1}^{(f)}(\mathbf{x}^*, \mathbf{x}^*) \boldsymbol{\Sigma}_{k,p,q_1}^{(W)}(\mathbf{x}^*, \mathbf{x}^*)\right]
\end{aligned}
\tag{47}
$$

In all other cases there will be no square terms

$$
\sum_{q_1=1}^{Q} \sum_{q_2 \neq q_1}^{Q} \mathbb{E}_{q_k(\mathbf{f}^*, \mathbf{W}^*)} \left[\mathbf{W}_{k,p,q_1}^* \mathbf{f}_{k,q_1}^* \mathbf{f}_{k,q_2}^* \mathbf{W}_{k,p,q_2}^*\right] = \sum_{q_1=1}^{Q} \sum_{q_2 \neq q_1}^{Q} \boldsymbol{\mu}_{k,p,q_1}^{(W)}(\mathbf{x}^*) \boldsymbol{\mu}_{k,q_1}^{(f)}(\mathbf{x}^*) \boldsymbol{\mu}_{k,q_2}^{(f)}(\mathbf{x}^*) \boldsymbol{\mu}_{k,p,q_2}^{(W)}(\mathbf{x}^*).
\tag{48}
$$

Combining both cases together we can write

$$\mathbb{E}\left[(\mathbf{y}_p^*)^2\right] = \sum_{k=1}^{K} \pi_k \sigma_{\mathbf{y}_p}^2 I + \sum_{k=1}^{K} \pi_k \sum_{q_1=1}^{Q} [\boldsymbol{\mu}_{k,p,q_1}^{(W)}(\mathbf{x}^*)\boldsymbol{\Sigma}_{k,q_1}^{(f)}(\mathbf{x}^*,\mathbf{x}^*)\boldsymbol{\mu}_{k,p,q_1}^{(W)}(\mathbf{x}^*)$$

$$+ \boldsymbol{\mu}_{k,q_1}^{(f)}(\mathbf{x}^*)\boldsymbol{\Sigma}_{k,p,q_1}^{(W)}(\mathbf{x}^*,\mathbf{x}^*)\boldsymbol{\mu}_{k,q_1}^{(f)}(\mathbf{x}^*) + \boldsymbol{\Sigma}_{k,q_1}^{(f)}(\mathbf{x}^*,\mathbf{x}^*)\boldsymbol{\Sigma}_{k,p,q_1}^{(W)}(\mathbf{x}^*,\mathbf{x}^*)] \qquad (49)$$

$$+ \sum_{k=1}^{K} \pi_k \sum_{q_1=1}^{Q} \sum_{q_2=1}^{Q} \boldsymbol{\mu}_{k,p,q_1}^{(W)}(\mathbf{x}^*)\boldsymbol{\mu}_{k,q_1}^{(f)}(\mathbf{x}^*)\boldsymbol{\mu}_{k,q_2}^{(f)}(\mathbf{x}^*)\boldsymbol{\mu}_{k,p,q_2}^{(W)}(\mathbf{x}^*).$$

# B  Synthetic Examples

Apart from the variational experiments in Section 2 of the main paper, additional experiments using a Markov Chain Monte Carlo (MCMC) approach are conducted in this section. We show that when the dependency structure is lost through the product likelihood construction, the mean of the posterior distribution for the latent function will also deviate from the true one. We also demonstrate the posterior contraction and the effect of the different corrections.

## B.1  Data Generating Process

We generate two synthetic observation processes from:

$$y_i^{(1)} = f(x_i) + \epsilon_1$$

$$y_j^{(2)} = \frac{1}{2} \sum_{k=2j-1}^{2j} y_k^{(1)} + \epsilon_2 \qquad (50)$$

Where $y_i^{(1)}, i = 1, ..., 2N$ is the observed value of $f(x_i)$ with noise $\epsilon_1 \sim \mathcal{N}(0, \sigma_1^2)$ and $y_j^{(2)}, j = 1, ..., N$ is the aggregate function of $y^{(1)}$ with noise $\epsilon_2 \sim \mathcal{N}(0, \sigma_2^2), \sigma_1 = 1, \sigma_2 = 0.1$. We are using a $\sin$ function to generate data $f(x_i) = 5\sin^2(x_i)$. The likelihood function with the observation processes $\mathbf{Y_1} = \{y_i^{(1)}\}_{i=1}^{2N}$, $\mathbf{Y_2} = \{y_j^{(2)}\}_{j=1}^{N}$ is given by:

$$L(\mathbf{Y}_1, \mathbf{Y}_2) = p(\mathbf{Y}_1|\mathbf{f}(\mathbf{x}), \sigma_1^2)p(\mathbf{Y}_2|\mathbf{Y}_1, \sigma_2^2) \qquad (51)$$

When the data from $\mathbf{Y}_1$ has the same support as the observation process $\mathbf{Y}_2$, the evidence from $\mathbf{Y}_2$ will not affect parameter estimation in the probability function $p(\mathbf{Y}_1|\mathbf{f}(\mathbf{x}), \sigma_1)$. However, when $\mathbf{Y}_2$ has different support from the observed $\mathbf{Y}_1$, the additional evidence should impact parameter inference. As $\mathbf{Y}_2$ does not depend on the latent function, this evidence will be hard to pass via the likelihood function in Eq. 51. One way to correct for this is to introduce dependency between $\mathbf{Y}_1$ and $\mathbf{Y}_2$ through a non-parametric prior over the latent function $\mathbf{f}(\mathbf{x})$.

## B.2  Gaussian Processes: Product Likelihood

Since the two observation processes follow the same underlying function $sin^2(x)$, we use a single Gaussian process to model the latent function $f(x)$. We assume:

$$f(x) \sim \mathcal{GP}(0, k(x, x')) \qquad (52)$$

where $k(x, x')$ is the covariance function of $f(x)$. We are using the squared exponential kernel:

$$k(x, x') = A\exp(-\frac{(x - x')^2}{l}) \qquad (53)$$

where $A$ is the amplitude parameter and $l$ is the length scale for the kernel function. Thus, we can write down the joint distribution of $\mathbf{Y}_1$ and $\mathbf{Y}_2$ as:

$$p(\mathbf{Y}_1, \mathbf{Y}_2, \mathbf{f}(\mathbf{x})) = p(\mathbf{f}(\mathbf{x})|\theta)p(\mathbf{Y}_1|\mathbf{f}(\mathbf{x}), \sigma_1^2)p(\mathbf{Y}_2|\mathbf{Y}_1, \sigma_2^2) \qquad (54)$$

Where $\theta$ is the hyper-parameters for the Gaussian process. We can write down the distribution of $\mathbf{Y}_1$ and $\mathbf{Y}_2$ by marginalizing out the latent function:

$$p(\mathbf{Y}_1, \mathbf{Y}_2) = \int p(\mathbf{f}(\mathbf{x})|\theta)p(\mathbf{Y}_1|\mathbf{f}(\mathbf{x}), \sigma_1^2)p(\mathbf{Y}_2|\mathbf{Y}_1, \sigma_2^2)d\mathbf{f}(\mathbf{x}) \tag{55}$$

As $\mathbf{Y}_1$ has all the information of $\mathbf{f}(\mathbf{x})$, this integral is tractable and we can write $y^{(1)} \sim \mathcal{GP}(0, k(x, x') + \sigma_1)$. But when the aggregation function $\mathbf{Y}_2$ has additional information about the latent function, i.e. $\mathbf{Y}_1$ and $\mathbf{Y}_2$ only partially overlapping, bringing additional information from $\mathbf{Y}_2$ requires the prediction of the missing values of the corresponding $\mathbf{Y}_1$ process. This can be done in Markov Chain Monte Carlo(MCMC) setting by treating the unobserved value of $\mathbf{Y}_1$ as extra parameters. However, this increase a lot of computational complexity for the MCMC sampler. One option is to make an independence assumption for $\mathbf{Y}_1$ and $\mathbf{Y}_2$. Thus, the information in $\mathbf{Y}_2$ can affect the latent function $\mathbf{f}$ directly.

### B.3 Gaussian Processes: Composite Likelihood

Using the composite likelihoods, we assume each part of the likelihood is independent to each other. For the joint probability of $\mathbf{Y}_1$ and $\mathbf{Y}_2$ , we have:

$$p(\mathbf{Y}_1, \mathbf{Y}_2) = \int p(\mathbf{f}(\mathbf{x})|\theta)p(\mathbf{Y}_1|\mathbf{f}(\mathbf{x}), \sigma_1^2)p(\mathbf{Y}_2|\mathbf{f}(\mathbf{x}), \sigma_2^2)d\mathbf{f}(\mathbf{x}) \tag{56}$$

Instead of assuming the conditional probability $p(\mathbf{Y}_2|\mathbf{Y}_1, \sigma_2^2)$, we are now assuming the data depends on the latent function $\mathbf{f}(\mathbf{x})$ directly. However, when $\mathbf{Y}_1$ and $\mathbf{Y}_2$ are different resolutions under the same support, this likelihood misspecifies the correlation and will make the inference of $f(x)$ contract into the observed mean. While this contraction actually equals to an extra bias to the data in the overlapping zone, the misspecified dependency structure will lead to an overfitting problem. This overfitting problem of product likelihoods has been studied in the information theory [31, 34] and the simplest way is to use an exponential weight to correct the inference:

$$L(\mathbf{Y}_1, \mathbf{Y}_2) = \int p(\mathbf{f}(\mathbf{x})|\theta)p(\mathbf{Y}_1|\mathbf{f}(\mathbf{x}), \sigma_1^2)^\alpha p(\mathbf{Y}_2|\mathbf{f}(\mathbf{x}), \sigma_2^2)^\alpha d\mathbf{f}(\mathbf{x}) \tag{57}$$

where $\alpha \in \mathbb{R}_{>0}$ is composite weight for the likelihood. The problem of learning the latent function becomes learning the parameters of the likelihood function and the composite weights.

### B.4 Composite Weights

The composite log likelihood function can be written as:

$$\ell_c(\hat{\theta}) = \sum_{i=1}^{k} f(\hat{\theta}_i|\mathbf{Y}) \tag{58}$$

where $f(\hat{\theta}_i|Y)$ is the likelihood function of $i$-th parameter $\theta_i$ and we assume each part of the likelihood function is independent to each other. $\hat{\theta}_i$ is the estimated value of $\theta_i$. With the observed distribution of $\mathbf{Y}$, $p_0(\mathbf{Y}|\theta_0)$ and $\theta_0$ as the true parameter value, we have:

$$\ell_c'(\theta_0) = \ell_c'(\hat{\theta}) + (\theta_0 - \hat{\theta})\ell_c''(\hat{\theta}) + o(n^{-1}) \tag{59}$$

$$\hat{\theta} - \theta_0 \to -\frac{\ell'(\theta_0)}{\ell''(\theta_0)} \tag{60}$$

Since we have $\ell'(\theta) = J(\theta)$ and $\ell''(\theta) = H(\theta)$, the variance of $\theta$ will follow the sandwich variance $H^{-1}(\theta)J(\theta)H^{-1}(\theta)$. Then, calculating the Taylor expansion for the likelihood, we have:

$$\ell_c(\theta_0) = \ell_c(\hat{\theta}) + (\theta_0 - \hat{\theta})\ell_c'(\hat{\theta}) + \frac{1}{2}(\theta_0 - \hat{\theta})\ell_c''(\hat{\theta})(\theta_0 - \hat{\theta})^T + o(n^{-1}) \tag{61}$$

The expected variance from the composite likelihood model is :

$$\mathbb{E}_\theta[Var(\hat{\theta}|\mathbf{Y})] = -H(\hat{\theta}|\mathbf{Y}) \tag{62}$$

Since $\hat{\theta} \to \theta$, we need to set the variance of the estimated parameter to the asymptotic variance. Thus, we have:

$$\mathbb{E}_\theta[\alpha Var(\hat{\theta}|\mathbf{Y})] = H^{-1}(\hat{\theta}|\mathbf{Y})J(\hat{\theta}|\mathbf{Y})H^{-1}(\hat{\theta}|\mathbf{Y}) \tag{63}$$

For a scalar variable, we can match the variance to the exact asymptotic vairance using a scalar number. But if the estimating variable $\theta$ is high dimensional, it's not easy to adjust the proper variance using a single weight. We could use a matrix ($\mathbf{C} \in \mathbb{R}^{k \times k}$) to adjust the covariance structure. In this case we would have:

$$\mathbf{C}H(\hat{\theta}|\mathbf{Y})\mathbf{C}^T = H^{-1}(\hat{\theta}|\mathbf{Y})J(\hat{\theta}|\mathbf{Y})H^{-1}(\hat{\theta}|\mathbf{Y}) \tag{64}$$

However, this increases the computational complexity substantially. One alternative way is to use a scalar weight to match the identities of the covariance matrix. Lyddon et al. [17] and Ribatet [27] developed two different ways to adjust the identities of the covariance matrix:

$$\alpha_{\text{Ribatet}} = \frac{|\hat{\theta}|}{Tr[\mathbf{H}(\hat{\theta})^{-1}\mathbf{J}(\hat{\theta})]} \quad , \quad \alpha_{\text{Lyddon}} = \frac{Tr[\mathbf{H}(\hat{\theta})\mathbf{J}(\hat{\theta})^{-1}\mathbf{H}(\hat{\theta})]}{Tr[\mathbf{H}(\hat{\theta})]}. \tag{65}$$

Where $\alpha_{\text{Ribatet}}$ considers all the information in the covariance matrix and $\alpha_{\text{Lyddon}}$ only matches the information in the diagonal elements.

## B.5 MCMC Composite Likelihood Experiments

We now construct an MCMC experiment for the synthetic data using Eq. 50. Instead of sampling directly from the intractable joint distribution of $L(\mathbf{Y_1}, \mathbf{Y_2})$, we sample from the joint probability with the latent variable $L(\mathbf{Y_1}, \mathbf{Y_2}, \mathbf{f(x)})$ via a Metropolis-Hastings within Gibbs sampler. We perform three block updates: on $\theta_0$ for the Gaussian process prior, $\mathbf{f(x)}$ for the latent function variables and $\sigma^2 = \{\sigma_1^2, \sigma_2^2\}$ for the noise parameter.

---

**Algorithm 3** Block Metropolis-Hastings within Gibbs

---

**Input:** Observed datasets $\{(\mathbf{X}_s, \mathbf{Y}_s)\}_{s=1}^S$, initial parameters $\theta_0$,

**for** $i$-th iteration **do**
    Update parameter block $\theta_i$
    **function** BLOCK $(\theta_i)$
        1. Sample proposed value of the Gaussian process prior $\theta_i' \sim N(\theta_{i-1}, \Delta)$

        2. Calculate the conditional probability distribution $p(\theta_i'|Y_1, Y_2, \sigma_{i-1}^2, f(x)_{i-1})$
        3. Calculate the acceptance rejection ratio:

    $\pi = \frac{p(\theta_i'|Y_1, Y_2, \sigma_{i-1}^2, \mathbf{f(x)}_{i-1})}{p(\theta_{i-1}|Y_1, Y_2, \sigma_{i-1}^2, \mathbf{f(x)}_{i-1})}$

        4. Update the $i$-th value of $\theta_i$ via $\pi$

    **end function**

    Update the parameter block $\mathbf{f(x)}_i$ via $\pi = \frac{p(\mathbf{f(x)}_i'|Y_1, Y_2, \sigma_{i-1}^2, \theta_i)}{p(\mathbf{f(x)}_{i-1}|Y_1, Y_2, \sigma_{i-1}^2, \theta_i)}$

    Update the parameter block $\sigma_i$ via $\pi = \frac{p(\sigma_i'^2|Y_1, Y_2, \theta_i, \mathbf{f(x)}_i)}{p(\sigma_{i-1}^2|Y_1, Y_2, \theta_i, \mathbf{f(x)}_i)}$

**end for**
    **return** $\theta, \mathbf{f(x)}, \sigma^2$

---

Figure 5: **Top Left:** Comparing the posterior of the latent function $f(x)$ under the product likelihood assumption and a correctly specified likelihood. The product likelihood assumption causes extreme posterior contraction which effects both the mean and variance. **Bottom Left:** Comparison of the true posterior of $\sigma_1$ noise to the posterior under the product likelihood and under the composite likelihood with different weights. The composite likelihood is able to recover the true posterior. **Top Right:** Comparing the posterior of the latent function $f(x)$ under the composite likelihood assumption with Lyddon correction and a correctly specified likelihood.**Bottom Right:** Comparing the posterior of the latent function $f(x)$ under the composite likelihood assumption with Ribatet correction and a correctly specified likelihood. The two correction have the similar results for our experiments. Although the mean of the function is not exactly match the true function, the variance of the latent function is corrected close to the true function. The misspecified part is due to the imprefect match of the asymptotic variance discussed in section B.4

## B.6 Variational Composite Likelihood Experiments

In this section we provide further details to reproduce the variational composite likelihood experiments in Sec. 2.2 and Fig. 2 of the main paper.

**Data Generation** We consider the case of having two dependent observation processes. We generate one process $\mathbf{Y}_1 = 5 \cdot \sin(\mathbf{X})^2 + 0.1 \cdot \epsilon$ with $\epsilon \sim \mathcal{N}(0,1)$ with 100 samples over the range $[-2, 15]$. For $\mathbf{Y}_2$ we aggregate $\mathbf{Y}_1$ into bins of size 3, $\mathcal{S}_2 = 3$, so that $\mathbf{Y}_2 \in \mathbb{R}^{33}$ and $\mathbf{X}_2 \in \mathbb{R}^{33 \times 3}$. In Fig. 2 we only plot the range $[3, 10]$.

**Parameter Initialization** For both MR-GP and VBAGG-NORMAL we use an SE kernels with length-scale of 0.1 and variance 1.0. We initialize the likelihood noise to 0.1.

**Additional Training Details** For VBAGG-NORMAL we run for 5000 epochs. For MR-GP for we run the MLE estimate for 5000 epochs, obtain $\alpha_{\text{Ribatet}}$ and then optimize the ELBO for 5000 epochs.

# C  Multi-resolution Air Pollution Experiments

## C.1  Inter-task Multi-resolution: $PM_{10}$-$PM_{25}$

In this section we provide additional details for reproducing the inter-task multi-resolution experiments as described in the main paper.

**Variational Parameter Initialization**: For MR-DGP we initialize all likelihood noises to 0.01 and we use a Matern32 kernel for all latent functions with a lengthscale of 0.01. For both MR-GPRN and CENTER-POINT we initialize the likelihood noise to be 0.1 and use a squared exponential kernel for all latent functions. We use $Q = 1$ and set the lengthscale of $\mathbf{f}$ to be 0.1 and the lengthscales of $\mathbf{W}$ to be 3.0.

**Training Details**: We train MR-DGP for a total of 2000 iterations. We train both MR-GPRN and CENTER-POINT for 2000 iterations each.

## C.2  Intra-task Multi-resolution: Space-time $NO_2$

In this section we provide additional details for reproducing the *intra*-task multi-resolution experiments described in Sec. 4 of the main paper.

**Data pre-processing**: We extract spatial features based on the London road network (OS Highways) [2] and land use (UKMap) [3]. OS Highways is a dataset of every road in London with information of the length, road classification (A Road, B Road, etc). UKMap is a dataset of polygons where each polygon represents a physical entity, e.g a building, a river, a park, etc. UKMap provides additional information such as the height of the buildings and the area of the parks and rivers. For each input location we construct a buffer of approximately 100m (a radius 0.001 degrees in SRID:4326). Within the buffer zone we calculate the average length of the A-roads, the average ratio between the width of the roads and height of buildings on the corresponding roads, and the total area of vegetation and water. We convert all time stamps into unix epochs and we standardize all features before training. To approximate the integral in the likelihood (Eq. 4 in main text) we discretize the area of each satellite based observation input into a 10 by 10 uniform grid of lat-lon points.

**MR-DGP Architecture**: For MR-DGP we use the architecture described on the right subfigure of Fig. 3 in the main paper where $\mathbf{X}_2, \mathbf{Y}_2$ corresponds to the LAQN dataset and $\mathbf{X}_1, \mathbf{Y}_1$ to the satellite dataset. We give the initialization of the specific latent functions below.

**Variational Parameter Initialization**: For MR-GPRN and VBAGG-NORMAL we use 400 inducing points for all latent functions. Both the inducing function values and the variances are randomly initialized between 0 and 1. For MR-DGP the latent functions $\mathbf{f}_{2,2}$ and $\mathbf{f}_{1,1}$ we place 200 inducing points and for $\mathbf{f}_2$ we use 100. For all models we initialize the inducing points locations with K-means.

**Model Parameter Initialization**: In all models and latent function withing, MR-GPRN, VBAGG-NORMAL and MR-DGP we use SE kernels initialized with lengthscales of 0.1 and SE variance to 1.0. We initialize the likelihood noise to be 0.1.

**Additional Training Details**: We train MR-DGP for a total of 9000 iterations. We train both MR-GPRN, VBAGG-NORMAL and CENTER-POINT for 10000 iterations each.

# D  MR-DGP

In this section we provide the complete derivation of the variational lower bound for MR-DGP.

## D.1 Specification of the prior

The full prior for MR-DGP is given by

$$p(\mathbf{F}, \mathbf{M}) = \left( \prod_{p=2}^{P} p(\mathbf{f}_p \mid \mathbf{m}_p) p(\mathbf{m}_p \mid \mathbf{f}_{1,p}, \{\mathbf{f}_{a,p}^{(2)}\}_{a=2}^{\mathcal{A}}) \right) \left( \prod_{p=1}^{P} p(\mathbf{f}_{1,p}) \prod_{a=2}^{\mathcal{A}} p(\mathbf{f}_{a,p}^{(2)} \mid \mathbf{f}_{a,p}) p(\mathbf{f}_{a,p}) \right)$$

(66)

We say that each of the GPs $\mathbf{f}_{a,p}$ are base GPs and to simplify notation in subsequent sections we use the function $\mathrm{Pa}(\cdot)$ to denote the set of parent functions for each node. For example, $\mathrm{Pa}(\mathbf{m}_2) = \{\mathbf{f}_{1,p}\} \cup \{\mathbf{f}_{a,p}^{(2)}\}$.

## D.2 Likelihood

The likelihood for MR-DGP (as illustrated in in Fig. 3) is

$$p(\mathbf{Y} \mid \mathbf{F}) = \prod_{p=2}^{P} p(\mathbf{Y}_{1,1} \mid \mathbf{f}_p) \prod_{p=1}^{P} p(\mathbf{Y}_{1,p} \mid \mathbf{f}_{a,p}) \prod_{a=2}^{\mathcal{A}} p(\mathbf{Y}_{1,p} \mid \mathbf{f}_{a,1}^{(2)}) p(\mathbf{Y}_{a,p} \mid \mathbf{f}_{a,p})$$

(67)

where each likelihood component is a multi-resolution likelihood of the form

$$\mathcal{N} \left( \mathbf{Y}_{a,n} \;\middle|\; \frac{1}{|\mathcal{S}_a|} \int_{S_{a,n}} \mathbf{f}_a^{(k)}(\mathbf{X}) \, d\mathbf{X}, \sigma_{a,k}^2 \mathbf{I} \right),$$

(68)

again we discretise the integral with a uniform grid over $S_a$.

## D.3 Augmented Prior

To allow for efficient inference we sparsify each GP by introducing inducing points:

$$\begin{aligned}
p(\mathbf{F}, \mathbf{M}, \mathbf{U}) =\; & p(\mathbf{F}, \mathbf{M}|\mathbf{U})p(\mathbf{U}) \\
=\; & \left( \prod_{p=2}^{P} p(\mathbf{f}_p|\mathbf{m}_p, \mathbf{u}_p) p(\mathbf{m}_p|\mathrm{Pa}(\mathbf{m}_p)) \right) \cdot \\
& \left( \prod_{p=1}^{P} p(\mathbf{f}_{1,p}|\mathbf{u}_{1,p}) \prod_{a=2}^{\mathcal{A}} p(\mathbf{f}_{a,p}^{(2)}|\mathbf{f}_{a,p}, \mathbf{u}_{a,p}^{(2)}) p(\mathbf{f}_{a,p}|\mathbf{u}_{a,p}) \right) \cdot \\
& \left( \prod_{p=2}^{P} p(\mathbf{u}_p) \cdot \prod_{p=1}^{P} p(\mathbf{u}_{1,1}) \prod_{a=2}^{\mathcal{A}} p(\mathbf{u}_{a,p}^{(2)}) p(\mathbf{u}_{a,p}) \right)
\end{aligned}$$

(69)

where each $p(\mathbf{u}_{\cdot,\cdot}^{(\cdot)}) = \mathcal{N}(\mathbf{u}_{\cdot,\cdot}^{(\cdot)} \mid 0, \mathbf{K}_{\cdot,\cdot}^{(\cdot)}(\mathbf{Z}_{\cdot,\cdot}^{(\cdot)}, \mathbf{Z}_{\cdot,\cdot}^{(\cdot)}))$ for $\mathbf{u}_{\cdot,\cdot}^{(\cdot)} \in \mathbb{R}^M$. The locations of the inducing points for the base GPs are $\mathbf{Z}_{a,p} \in \mathbb{R}^{M \times D}$ and for the deep GPs $\mathbf{Z}_p, \mathbf{Z}_{a,p}^2 \in \mathbb{R}^{M \times D}$. For brevity we have omitted the conditional on the inducing locations $\mathbf{Z}$ in our notation.

## D.4 Variational approximate Posterior

Following [28] we construct an approximate augmented posterior that maintains the dependency structure between layers:

$$\begin{aligned}
q(\mathbf{M}, \mathbf{F}, \mathbf{U}) =\; & p(\mathbf{M}, \mathbf{F}|\mathbf{U})q(\mathbf{U}) \\
=\; & p(\mathbf{M}, \mathbf{F}|\mathbf{U}) \prod_{p=2}^{P} q(\mathbf{u}_p) \cdot \prod_{p=1}^{P} q(\mathbf{u}_{1,1}) \prod_{a=1}^{\mathcal{A}} q(\mathbf{u}_{a,p}^{(2)}) q(\mathbf{u}_{a,p})
\end{aligned}$$

(70)

where each $q(\mathbf{u}_{\cdot,\cdot}^{(\cdot)})$ are standard free-form Gaussians $\mathcal{N}(\mathbf{u}_{\cdot,\cdot}^{(\cdot)} \mid \mathbf{m}_{\cdot,\cdot}^{(\cdot)}, \mathbf{S}_{\cdot,\cdot}^{(\cdot)})$. The conditional is of the form

$$p(\mathbf{f}_{a,p}|\mathrm{Pa}(\mathbf{f}_{a,p}), \mathbf{u}_{a,p}) = \mathcal{N}(\mathbf{f}_{a,p} \mid \mu'_{a,p}, \Sigma'_{a,p}) \tag{71}$$

with mean and variance given by the standard conditional equations

$$\begin{aligned}
\mu'_{a,p} &= \alpha_{a,p} \mathbf{K}_{a,p}(\mathbf{Z}_{a,p}, \mathbf{Z}_{a,p})^{-1} \mathbf{u}_{a,p}, \\
\Sigma'_{a,p} &= \mathbf{K}_{a,p}(\mathrm{Pa}(\mathbf{f}_{a,p}), \mathrm{Pa}(\mathbf{f}_{a,p})) - \alpha_{a,p} \mathbf{K}_{a,p}(\mathbf{Z}_{a,p}, \mathbf{Z}_{a,p})^{-1} \alpha_{a,p}^T.
\end{aligned} \tag{72}$$

where $\alpha_{a,p} = \mathbf{K}_{a,p}(\mathrm{Pa}(\mathbf{f}_{a,p}), \mathbf{Z}_{a,p})$.

### D.5 Marginalisation over inducing points

Firstly we can marginalise the inducing variables analytically

$$\begin{aligned}
q(\{\mathbf{m}_p\}_p^P, \{\mathbf{f}_{a,p}\}_{a,p=1}^{\mathcal{A},P}) &= \int q(\{\mathbf{m}_p\}_p^P, \{\mathbf{f}_{a,p}, \mathbf{u}_{a,p}\}_{a,p=1}^{\mathcal{A},P}) \, d\mathbf{u}_{1,1} \cdots d\mathbf{u}_{\mathcal{A},P} \\
&= \prod_{p=1}^P p(\mathbf{m}_p|\mathrm{Pa}(\mathbf{m}_p)) \prod_{a=1}^{\mathcal{A}} \int p(\mathbf{f}_{a,p}|\mathrm{Pa}(\mathbf{f}_{a,p}), \mathbf{u}_{a,p}) q(\mathbf{u}_{a,p}) \, d\mathbf{u}_{a,p} \\
&= \prod_{p=1}^P p(\mathbf{m}_p|\mathrm{Pa}(\mathbf{m}_p)) \prod_{a=1}^{\mathcal{A}} q(\mathbf{f}_{a,p}|\mathrm{Pa}(\mathbf{f}_{a,p}))
\end{aligned} \tag{73}$$

The integral can evaluated in closed form resulting in

$$q(\mathbf{f}_{a,p}|\mathrm{Pa}(\mathbf{f}_{a,p})) = \mathcal{N}(\mathbf{f}_{a,p} \mid \mu_{a,p}, \Sigma_{a,p}) \tag{74}$$

where the mean and variance are given by

$$\begin{aligned}
\mu_{a,p} &= \alpha_{a,p} \mathbf{K}_{a,p}(\mathbf{Z}_{a,p}, \mathbf{Z}_{a,p})^{-1} \mathbf{m}_{a,p}, \\
\Sigma_{a,p} &= \Sigma'_{a,p} - \alpha_{a,p} \mathbf{K}_{a,p}(\mathbf{Z}_{a,p}, \mathbf{Z}_{a,p})^{-1} \mathbf{S}_{a,p} \mathbf{K}_{a,p}(\mathbf{Z}_{a,p}, \mathbf{Z}_{a,p})^{-1} \alpha_{a,p}^T
\end{aligned} \tag{75}$$

### D.6 Mixture of Experts

We define $p(\mathbf{m}_p|\mathrm{Pa}(\mathbf{m}_p))$ as a mixture of experts, that is a weighted combination of the local experts

$$p(\mathbf{m}_p|\mathrm{Pa}(\mathbf{m}_p)) = \mathcal{N}\left(\mathbf{m}_p \;\middle|\; \sum_a^{\mathcal{A}} \mathbf{w}_{a,p}\mu_{a,p}, \sum_a^{\mathcal{A}} \mathbf{w}_{a,p}\Sigma_{a,p}\mathbf{w}_{a,p}\right). \tag{76}$$

The weights $\mathbf{w}_{a,p}$ are application specific and we provide specific examples in Section D.9.

### D.7 Marginalisation over layers

We follow the doubly stochastic framework of [28] and marginalise through the layers using Monte Carlo estimates. The first layer of GPs can be sampled from directly, these samples are then propagated through all the subsequent layers:

$$q(\mathbf{f}_{a,p}) = \int q(\mathbf{f}_{a,p}|\mathrm{Pa}(\mathbf{f}_{a,p})) \prod_{l=1}^{\mathcal{L}-1} q(\mathrm{Pa}^{(l)}(\mathbf{f}_{a,p})|\mathrm{Pa}^{(l+1)}(\mathbf{f}_{a,p})) \, d\mathrm{Pa}^{(1)}(\mathbf{f}_{a,p}) \cdots d\mathrm{Pa}^{(\mathcal{L})}(\mathbf{f}_{a,p}), \tag{77}$$

where each $q(\mathbf{f}_{a,p}|\mathrm{Pa}(\mathbf{f}_{a,p}))$ is a DGP of the form Eqn. 6.

## D.8 Variational Lower Bound

In this section we provide the derivation of the variational lower bound for MR-DGP. The evidence lower bound (ELBO), which lower bounds the log marginal likelihood $\log p(\mathbf{Y}|\mathbf{X})$, is

$$\mathcal{L} = \mathbb{E}_{q(\mathbf{M},\mathbf{F},\mathbf{U})}\left[\log\frac{p(\mathbf{Y},\mathbf{M},\mathbf{F},\mathbf{U})}{q(\mathbf{M},\mathbf{F},\mathbf{U})}\right] = \mathbb{E}_{q(\mathbf{M},\mathbf{F},\mathbf{U})}\left[\log\frac{p(\mathbf{Y}|\mathbf{F})p(\mathbf{M},\mathbf{F}|\mathbf{U})p(\mathbf{U})}{p(\mathbf{M},\mathbf{F}|\mathbf{U})q(\mathbf{U})}\right] \tag{78}$$

Cancelling the relevant terms inside the logarithm we get

$$\mathcal{L}_{\text{MR-DGP}} = \underbrace{\mathbb{E}_{q(\mathbf{M},\mathbf{F},\mathbf{U})}\left[\log p(\mathbf{Y}|\mathbf{F})\right]}_{\text{ELL}} + \underbrace{\mathbb{E}_{q(\mathbf{U})}\left[\log\frac{P(\mathbf{U})}{q(\mathbf{U})}\right]}_{\text{KL}}.$$

Note that we have slightly abused notation to keep the derivation clear and now provide the full expanded lower bound:

$$\text{ELL} = \sum_{p=2}^{P}\mathbb{E}_{q(\mathbf{f}_p)}\left[\log p(\mathbf{Y}_{1,1}|\mathbf{f}_p)\right] + \sum_{p=1}^{P}\sum_{a}^{\mathcal{A}}\left[\mathbb{E}_{q(\mathbf{f}_{\mathbf{a},\mathbf{1}}^{(\mathbf{2})})}\left[p(\mathbf{Y}_{1,p}|\mathbf{f}_{\mathbf{a},\mathbf{1}}^{(\mathbf{2})})\right] + \mathbb{E}_{q(\mathbf{f}_{a,p})}\left[p(\mathbf{Y}_{a,p}|\mathbf{f}_{a,p})\right]\right] \tag{79}$$

and

$$\text{KL} = \mathbb{E}_{q(\mathbf{U})}\left[\log\frac{\left(\prod_{p=2}^{P}p(\mathbf{u}_p)\cdot\prod_{p=1}^{P}p(\mathbf{u}_{1,1})\prod_{a=2}^{\mathcal{A}}p(\mathbf{u}_{a,p}^{(2)})p(\mathbf{u}_{a,p})\right)}{\left(\prod_{p=2}^{P}q(\mathbf{u}_p)\cdot\prod_{p=1}^{P}q(\mathbf{u}_{1,1})\prod_{a=2}^{\mathcal{A}}q(\mathbf{u}_{a,p}^{(2)})q(\mathbf{u}_{a,p})\right)}\right]. \tag{80}$$

The KL term can be computed in closed form because it is just the sum of KL terms between two Gaussians. The ELL term is approximated using the Monte Carlo estimates from marginalising through the layers.

## D.9 Mixture of Experts Weights

In this section we provide a specific and intuitive example of weights used when combining the mixture of experts. We derive weights that naturally weigh the experts by the level of support provided. We assume that this is defined by the resolution of the base layers, where higher resolutions are 'trusted' more. We first derive the weights for two generic GPs $\mathbf{f}_1$ and $\mathbf{f}_2$ and will generalise and apply the weights to the MR-GPRN after.

To find the support of an expert we utilise the differential entropy of the GP (see [13]):

$$H(p(\mathbf{f}|\mathbf{D})) = \frac{1}{2}\log(\sigma_{\mathbf{f}|\mathbf{D}}^2) + \frac{1}{2}(\log(2\pi) + 1). \tag{81}$$

For our mixing weights we are not interested in the amount of information each GP has, just whether there is any. So we drop the 2nd term on the rhs of Eq. 81, which we denote by $I(p(\mathbf{f}|\mathbf{D}))$, and normalise the information to be between zero and one. One such function is simply:

$$\hat{I}(\mathbf{f}) = \frac{I(p(\mathbf{f}|D)) - \min(I(p(\mathbf{f}|D)))}{\max(I(p(\mathbf{f}|D))) - \min(I(p(\mathbf{f}|D)))} \tag{82}$$

This is not the only normalisation that can be done, one could also use a sigmoid function such as $\tanh(\cdot)$. Now we want a function that weighs $\mathbf{f}_2$ down when $\mathbf{f}_1$ has information, inherently capturing that we trust $\mathbf{f}_1$ over $\mathbf{f}_2$. One such function is the normalised information of $\mathbf{f}_2$ minus the joint information of $\mathbf{f}_1$ and $\mathbf{f}_2$:

$$\hat{I}(\mathbf{f}_1) + \beta_2\hat{I}(\mathbf{f}_2) = \hat{I}(\mathbf{f}_1) + \hat{I}(\mathbf{f}_2) - \hat{I}(\mathbf{f}_1,\mathbf{f}_2). \tag{83}$$

Because we have normalised the information to be within zero and one we approximate the the joint information with a hadamard product (approximating an XOR function where the value is one if they both have information else it is zero).

$$\beta_2 \hat{I}(\mathbf{f}_2) = \hat{I}(\mathbf{f}_2) - \hat{I}(\mathbf{f}_1)\hat{I}(\mathbf{f}_2)$$
$$= (1 - \hat{I}(\mathbf{f}_1)). \tag{84}$$

The function $\beta_2$ now has maximum and minimum values in the range $[0, 1]$ and so we can directly use $\beta_2$ as our mixing weights. To makes all the weights sum to one we define

$$\beta_1 = 1 - \beta_2 = \hat{I}(\mathbf{f}_1). \tag{85}$$

### D.10 Combining arbitrary number of experts

We combine them in a hierarchical manner, in a similar way to [? ], and construct a computational graph. Given $P$ experts, the mixing weights are defined as

$$\mathbf{m} = \hat{I}(\mathbf{f}_1)\mathbf{f}_1 + (1 - \hat{I}(\mathbf{f}_1))(\hat{I}(\mathbf{f}_2)\mathbf{f_2} + (1 - \hat{I}(\mathbf{f}_2))(\cdots + (1 - \hat{I}(\mathbf{f}_{P-1}))\mathbf{f}_P)) \tag{86}$$

## E Relation to VBAgg

In this section we show that MR-GPRN can be seen as a generalisation of VBAGG-NORMAL [15] from a single GP to a GPRN. In VBAgg each observation $y^a$ is the aggregate output of some bag of items $\mathbf{x}^a = \{\mathbf{x}_i^a\}_{i=1}^{N_a}$. The likelihood of each bag is $y^a|\mathbf{x}^a \sim \mathcal{N}(y|\eta^a, \tau^a)$ where $\eta^a = \sum_{i=1}^{N_a} w_i^a \mu(\mathbf{x}_i^a)$ and $\mu$ is the mean of the latent process $f$. In MR-GPRN we are modelling the underlying process with the sum of products of GPs. Rewriting MR-GPRN using the notation of [15]: $\mu = \mathbf{W}\mathbf{f}$ and each dataset $\{\mathbf{X}_a, \mathbf{Y}_a\}_{a=1}^{\mathcal{A}}$ directly corresponds to the observations and bag of items defined in VBAGG-NORMAL. Let $N_a = \mathcal{S}_a$, and $\tau^a = \sigma_a^2$ and the composite weight $\alpha = 1$. The composite weight of value 1 is implicitly included in the model of VBAGG-NORMAL through the independence assumption. We assume an simple aggregation of the bag of items, although we note that is not necessary, so setting $w_i^a = \frac{1}{|\mathcal{S}_a|}$ we obtain $y^a \sim \mathcal{N}(\sum_{i=1}^{N_a} w_i^a \mu(\mathbf{x}_i^a), \tau^a)$ which is MR-GPRN in the notation of [15]. VBAGG-NORMAL is then recovered when we use only one latent function (by setting $\mathbf{W}$ to a constant value), by only considering the single task setting and by setting the composite weight to one.

The VBAgg model is defined by

$$y^a \sim p(y^a|\eta^a), \quad \eta^a = \sum_{i=1}^{N_a} w_i^a \eta_i^a = \sum_{i=1}^{N_a} w_i^a \Psi(f(x_i^a)) \tag{87}$$

where $y^a$ is the independent aggregate observations, $w_i^a$ are fixed weights and $f$ follows a Gaussian process. For the unobserved latent variables $y_i^a$, we assume:

$$y_i^a \sim p(y_i^a|\eta_i^a), \quad \eta_i^a = \Psi(f(x_i^a)) \tag{88}$$

When the probabilities of $y^a \sim p(y^a|\eta^a)$ and $y_i^a \sim p(y_i^a|\eta_i^a)$ are Gaussian, the model is equivalent to VBAGG-NORMAL. We firstly consider the special case, when $\Psi(f(x_i^a)) = f(x_i^a)$. As $f(x_i^a) \sim \mathcal{GP}(0, K(x_i^a, \cdot))$, we could say that $y_i^a \sim N(f(x_i^a), \sigma_i^a)$ still follows a Gaussian process and $w_i y_i^a \sim \mathcal{GP}(0, w_i^2 K(x_i^a, \cdot))$ with fixed weight $w_i$. Thus, due to the additive property of Gaussian processes, the aggregation function $y^a \sim \mathcal{GP}(0, \sum_i \sum w_i w_. K(x_i^a, \cdot))$. The covariance of different aggregated

observations is:

$$
\begin{aligned}
Cov(y^a, y^b) &= Cov\left(\sum_{i=1}^{N_a} w_i^a y_i^a, \sum_{j=1}^{N_b} w_j^b y_j^b\right) \\
&= \mathbb{E}\left[\sum_{i=1}^{N_a}\sum_{j=1}^{N_b} w_i^a y_i^a w_j^b y_j^b\right] - \mathbb{E}\left[\sum_{i=1}^{N_a} w_i^a y_i^a\right]\mathbb{E}\left[\sum_{j=1}^{N_b} w_j^b y_j^b\right] \\
&= \mathbb{E}\left[\sum_{i=1}^{N_a}\sum_{j=1}^{N_b} w_i^a y_i^a w_j^b y_j^b\right] \\
&= \sum_{i=1}^{N_a}\sum_{j=1}^{N_b} w_i^a w_j^b K(x_i^a, x_j^b) \\
&= \widetilde{K}(\mathbf{x}^a, \mathbf{x}^b)
\end{aligned}
\tag{89}
$$

Also, the covariance between $y_i^* = f(x_i^*)$ and $y^a$ is given:

$$
\begin{aligned}
Cov(y_i^*, y^a) &= Cov\left(y_i^*, \sum_{j=1}^{N_a} w_j^a y_j^a\right) \\
&= \sum_{j=1}^{N_a} w_j^a K(x_i^*, x_j^a) \\
&= \widetilde{K^*}(x_i^*, \mathbf{x}^a)
\end{aligned}
\tag{90}
$$

Thus, the problem of learning $f(\cdot)$ function becomes a standard Gaussian process regression problem and we have:

$$
\begin{bmatrix} \mathbf{y} \\ y_i^* \end{bmatrix} \sim N\left(0, \begin{bmatrix} \widetilde{K}(\mathbf{x}, \mathbf{x}) + \sigma_y^2\mathbf{I} & \widetilde{K^*}(x_i^*, \mathbf{x}) \\ \widetilde{K^*}(\mathbf{x}, x_i^*) & K(x_i^*, x_i^*) \end{bmatrix}\right)
\tag{91}
$$

where $\mathbf{y} = \{y^1, ..., y^a, ..., y^n\}$ and the predictive posterior is given by:

$$
\begin{aligned}
y_i^*|\mathbf{y}, \mathbf{x}, x_i^* &\sim N(\mu^*, \sigma^*) \\
\mu* &= \widetilde{K^*}(x_i^*, \mathbf{x})[\widetilde{K}(\mathbf{x}, \mathbf{x}) + \sigma_y^2\mathbf{I}]^{-1}\mathbf{y} \\
\sigma^* &= K(x_i^*, x_i^*) - \widetilde{K^*}(x_i^*, \mathbf{x})[\widetilde{K}(\mathbf{x}, \mathbf{x}) + \sigma_y^2\mathbf{I}]^{-1}\widetilde{K^*}(\mathbf{x}, x_i^*))
\end{aligned}
\tag{92}
$$

For a given generalized linear function $\Psi(f(x_i^a))$, we have $\Psi(\mathbf{f}(\mathbf{x}^a)) = \mathbf{W}_\Psi^a \mathbf{f}(\mathbf{x}^a)$, where $\mathbf{W}_\Psi^a$ is a $N_a \times N_a$ weighting matrix. When $\mathbf{f}(\mathbf{x}^a) \sim \mathcal{GP}(0, K(x_i^a, x_j^a))$, then $\Psi(\mathbf{f}(\mathbf{x}^a)) = \mathbf{W}_\Psi^a \mathbf{f}(\mathbf{x}^a) \sim \mathcal{GP}(0, \mathbf{W}_\Psi^a K(x_i^a, x_j^a)(\mathbf{W}_\Psi^a)^T))$ Thus, we have:

$$
\begin{aligned}
y^a &= \mathbf{w}^a[\Psi(\mathbf{f}(\mathbf{x}^a))]^T \\
&= \mathbf{w}^a[\mathbf{W}_\Psi^a \mathbf{f}(\mathbf{x}^a)]^T \sim \mathcal{GP}(0, \mathbf{w}^a[\mathbf{W}_\Psi^a \mathbf{K}(\mathbf{x}^a, \mathbf{x}^a)(\mathbf{W}_\Psi^a)^T](\mathbf{w}^a)^T)
\end{aligned}
\tag{93}
$$

Following the Eq. 89, the covariance is given by:

$$
Cov(y^a, y^b) = Cov(\mathbf{w}^a[\mathbf{W}_\Psi^a \mathbf{f}(\mathbf{x}^a)]^T, \mathbf{w}^b[\mathbf{W}_\Psi^b \mathbf{f}(\mathbf{x}^b)]^T) \sim \mathcal{GP}(0, \mathbf{w}^a[\mathbf{W}_\Psi^a \mathbf{K}(\mathbf{x}^a, \mathbf{x}^b)(\mathbf{W}_\Psi^b)^T](\mathbf{w}^b)^T)
\tag{94}
$$

When the function $\Psi(\cdot)$ is a fixed function and $\mathbf{W}_\Psi$ are constants we recover the VBAGG-NORMAL model. When the function $\Psi(\cdot)$ is a random function and $\mathbf{W}_\Psi$ are Gaussian processes, then $\Psi(\mathbf{f}(\mathbf{x}^a))$ is a GPRN and the aggregation function $y^a$ follows the MR-GPRN model without composite likelihood corrections.