[Reviews · NeurIPS 2019]

Reviewer 1



General comments and questions (mainly related to quality/clarity; for originality/significance; see elsewhere in the review) - Overall, I enjoyed reading the paper, but the paper is very technical and dense with many inline equations which makes the paper difficult to appreciate the first time around. I appreciate that these are needed to reproduce the paper. - The technical contributions appear sound and non-trivial. - The rationale and justification for the composite likelihood should be improved in the main text for the reader to appreciate line 86-96. Should $phi$ be included in Fig 2? Is $phi$ fixed once estimated or threated as a hyperparameter in the subsequent learning (and why)? Is the composite likelihood still part of the MR-DGP model (and why)? - The experiments are relevant and appear well-executed. If time permits, I think a very basic baseline would be appreciated, e.g. a very basic GP. For comparison, I think it would also be of relevance to include results for the MR-GPRN without the composite likelihood for the real-world dataset, Tab. 2). - The paper argues (line 62-63) that it is looking for large scale and quality uncertainty quantification. Mentioning the dataset size and perhaps reporting predictive log-likelihood for the real-world experiment would be appreciated to provide better evidence that has been achieved. - The std deviation on the (MSE and MAPE) predictions appears rather large; I think it would be helpful to explain/comment on this...?. Minor: - Check that $\theta$ defined in the main text. - Check that MAPE is clearly defined.

Reviewer 2



-- Paper Summary -- This paper uses a practical real-world problem involving measuring air pollution across London in order to frame two important modelling aspects for multi-task Gaussian process models. The first contribution draws from the literature on GPRNs for combining latent functions in multi-output problems so as to cater for aggregated outputs while also including composite likelihoods. The second primary contribution of the paper goes a step further by tackling possible intra-task dependencies arising by way of outputs for the same auxiliary task observed with increasing fidelity or resolution. The experiments compare the proposed MR-GPRN model, and its deep counterpart (MR-DGP), against a baseline method and recently published work by Law et al. (2018) on developing scalable GPs for aggregate count data. The evaluation indicates that MR-DGP outperforms the aforementioned techniques, while also exposing weaknesses in these methods (Figure 2). -- Originality and Significance -- Although the contributions of this work are predominantly framed within the real-world case study featured throughout the paper, there are indeed several interesting contributions being presented here, especially in relation to recent work by Law et al. (2018), Moreno-Munoz et al. (2018), and Cutajar et al. (2019). In many ways, the model presented here unifies several features of the aforementioned work, while also presenting a more general framework for modelling complex relationships between tasks which may also require more intricate intra-modelling. To this end, I think that the model being proposed in this paper should be of particular interest to practitioners relying on Gaussian processes for their work. I find the first segment of the paper to be particularly interesting. Perhaps the distinction to other work could be made clearer, however, especially in relation to the work of Law et al. (the section provided in the supplementary is great, but part of it should be included in the main paper in favour of other less essential material), and the variation of AutoGP by Krauth et al. (2017) which places a GPRN likelihood in a GP for multi-output problems. On the contrary, I must admit to feeling somewhat ambivalent about the DGP aspect of the paper. Although I see how the multi-resolution requirement arises from the real-world problem being considered in this work, I feel as though it adds very little to the more interesting theoretical aspects of the paper featured in Section 3. The resulting graphical model is fairly similar to that presented in Cutajar et al. (2019), sans the mixture of experts approached championed for here. Perhaps because it is not the primary scope of the paper, there is currently little information on the choice of kernel for the intermediate-layer GPs combining low-resolution outputs and input points from the actual domain. This could perhaps be confusing to an audience who isn’t familiar with such multi-fidelity models. Either way, making the distinction between multi-fidelity and multi-resolution more explicit would further clarify the contributions of this paper. On a related note, in its current format, the discussion on DGPs comes across as overly cumbersome. Given that the discussion on inducing points for MRDGP appears to be fairly standard, I believe this can be shortened or moved to the supplementary in order to reduce clutter in the main text. Tangentially, I don’t fully understand the emphasis on how the method ‘learn[s] a forward mapping between observation processes’ as opposed to ‘trusting and propagating the mean’[L191/192]. Perhaps explicitly re-emphasising how this relates to the individual bias associated with intermediate layers could be helpful here. -- Technical Quality/Evaluation -- The formulation of the model is thoroughly explained throughout the paper, and it is clear from the extensive and rigorous supplementary material that the authors made an effort in ensuring that all results can be re-derived. The theoretical aspect of the paper appears to be correct, although I admit to having only briefly skimmed through the supplementary material. The implementation of the model as an extension to GPflow should also encourage greater reproducibility and extendibility due to widespread familiarity with the library. The evaluation is interesting since it teases apart a very complex real-world problem, while also highlighting individual aspects of the model by way of synthetic experiments (featured in both the main paper and the supplement). Overall, this appears to be suitably comprehensive and gives credence to the practical appeal of the method, whereby the performance gains are quite substantial. -- Writing/Clarity -- Overall, this is a well-written paper with a clearly motivated problem statement. I liked that the initial example describing pollution sensors in London is developed from the introduction through to the end, as this effectively showcases the contributions featured in the paper in a consistent manner. The graphical models and algorithm snippets are also clear and helpful, and are appropriately tied to the main text. A few comments on the paper’s presentation: - Some minor typos/preferences: L29L: I would change the last part of the sentence to ‘… contraction, as well as degradation of performance and uncertainty calibration.’; L44: Related work precedes the experiments; L70: task $p$; L75: model directly -> directly model; L75: , but it; Eqn1: Wrong punctuation; Eqn2: Wrong punctuation; L109: I don’t think $R_a$ has been used elsewhere?; L128: ‘weigh higher’ -> ‘assign more weight’; Algorithm2: Italicise ‘l’ in method signature; L180: missing citation; L185/193/197/203: Strange punctuation; - Some references are inconsistent for the same venue, e.g. [12] and [16]; - The use of \textsc{} and \textit{} is overly abundant, and makes the paper looks messy at times; - Having so many inline equations sometimes makes for difficult reading and cluttered presentation. Judging by the length of the supplementary material, I appreciate that the authors have already had to cut a fair amount of content from the paper. However, I believe there is still some fine-tuning to be done by shifting other well-known or easily-derived results to the supplementary. -- Overall recommendation -- In spite of some minor issues and lack of finesse in its presentation, I consider this to be a solid paper overall. However, my primary concern is that the authors tried to fit in too much content instead of judiciously expanding upon and highlighting more significant contributions. One of my suggestions would be to reduce the space which is currently dedicated to explaining the deep aspect of the model, which doesn’t deviate too far from related work and consequently doesn’t add much nuance to the discussion. On the contrary, I personally think that the focus of the paper should be placed on the components targeting inter-dependencies between processes, with the capability to handle intra-dependencies treated as an additional nice-to-have feature. P.S. A paper having similar goals appeared on arXiv shortly following the NeurIPS deadline: “Multi-task Learning for Aggregated Data using Gaussian Processes”, by Yousefi et al. In order to judge this submission fairly, I did not draw any comparisons to this paper in my review. Optionally, the authors may choose to briefly comment on the similarities/differences to this work. ** Post-rebuttal Update ** Thank you for your rebuttal. My opinion of the paper remains fairly similar to that expressed in my original review. I think it can definitely benefit from further refinement in the exposition of the various model components, and the experiments can also be broadened beyond the recurring case study given in the paper. Including more synthetic examples such as Figure 1 in the supplementary material could be a step in the right direction. Addressing the remarks made by Reviewer4 regarding the weights by including the extra experiments proposed in the rebuttal (L33) should also help alleviate these concerns, while also clarifying the composite likelihood aspect of the model. Overall, I think that the overall contributions of the paper are varied and interesting, although having more time to improve it further may work in its favour. My vote still tends towards acceptance, but I do not feel inclined to make a strong case for it.

Reviewer 3



*** Update after the rebuttal *** I believe the authors have provided a strong rebuttal, including a table with new results. However, the composite likelihood approach (CL) is a substantial part of the paper and I'm very concerned about whether it helps at modelling dependencies at all. There is a single composite weight in Equation 1 in the main paper, which comes out of the ELL term in Equation 6 in the supplement, i.e. all the resolutions have the same weight. This weight is fixed (using the proposed estimate) before carrying out variational inference. This means that all the components in the ELL term are weighted equally, hence I do not understand the claim that this is a way to "model the additional dependency of the observation processes". In the new table of results, MR-GPRN w/o CL and MR-GPRN w/ CL are essentially the same, which further supports my point about the CL approach not providing any additional benefit. The authors will need to discuss this as (in my view) the proposed CL+VI approach for modelling additional dependencies is either wrong or counterintuitive. **** ### Summary ### This paper develops Gaussian process-based methods for multi-resolution multi-task regression problems, i.e. problems with data collected at multiple resolutions and also with multiple outputs. (i) The paper proposes a flat GP method that builds upon the Gaussian process regression networks (GPRNs) of Wilson et al [30] by extending it to handle multiple resolutions through a composite likelihood model. (ii) The paper also develops a deep GP method based on GPRNs and mixture of experts. Experiments are presented on synthetic data and one real dataset concerning the estimation of air pollution levels in London. ### Originality ### The paper is mainly a combination of existing ideas, with regards to multi-output problems (using the GPRN of Wilson et al [3]), modelling dependencies in the likelihood (using a composite likelihood approach) and flexible and efficient modelling and inference for GPs and deep GPs [11, 24]. I do like the exploration of composite likelihood approaches for modelling dependencies across different modalities (e.g. resolution, outputs, etc). I have seen this before for structured likelihood settings, e.g. when using a pseudo-likelihood approach. However, the paper falls short in demonstrating that such approaches are worthwhile when compared with the natural approach of using latent variables. In other words, the benefits of the composite likelihood approach are unclear as a comparison with a baseline that uses the same GPRN-type model but without a composite likelihood is not provided. See more on baselines below. ### Quality ### - I believe the paper is technically sound, although there are some details that make the reader wonder whether there is something inherently wrong or the confusion may be caused by a lack of clarity or typos. For example, in Equation (1) and algorithm 1, only a single composite weight $\phi$ is used. I am not very familiar with the literature cited here but it looks like a single composite weight would not make a difference. One way to think about it is that this weight will come out of the expectation across multiple resolutions (during variational inference) and provide nothing with regards to modelling dependencies. Can please the authors clarify/explain what is going on? - Another reason (perhaps related to the above) for concern is the baseline used. My understanding of VBAgg [12] is that it is a method proposed for handling aggregate data, for example when the inputs are observed at a higher granularity than the output, and it is not really a multi-resolution method. A better baseline would be to consider a GPRN-type model where the resolution is seen as another context and where the GPRN parameters (e.g. weights) are now tensors. This would really help us understand what the composite likelihood is bringing to the game of modelling dependencies across resolutions. This is perhaps what the method is doing so it would probably materialize to not having the composite weights. In any case, the paper does need a multi-resolution baseline so it is unclear why there is no comparison with this (e.g. using [6] and simple extensions to multi-task). Similarly, other multi-task methods can be evaluated. - Additionally, with regards to the composite weights, is there any theoretical justification to first estimating the GPRN parameters using MLE so as to estimate the composite weights, and then fixing these to estimate the posterior of GPRN parameters using variational inference? It seems rather ad-hoc. Why not estimating everything under a single (e.g. variational) framework? - I see some inverse Hessians in the algorithm yet the computational complexity reported does not seem to include this. Can the authors please clarify? - Only point prediction metrics are used. Please report (estimates of) test likelihoods. ### Clarity ### - The paper is well structured but, at times, it becomes really difficult to follow what's going on. This is particularly true for the DGP model (section 4). I strongly suggest to the authors to pass this trough someone else (with a background in GPs) and they will realize I am probably right. For example, in line 131 it is very unclear what the GP is on as it seems that the kernel is computed on two different spaces? P(X_a) is not even defined. Why the notation P_a,k^{(1)} if the (1) is not used for anything? - Table 2 right: It shows that actually VBAgg is the best when using MAPE but the proposed method MR_DGP is bold. why? This is also not even discussed in the text. - The abstract needs some work: it refers to shallow GP mixtures. GPRNs are not really mixtures in the probabilistic sense. It is unclear what "naturally handle biases in the mean" means, also mentioned in the text. line 8, information-theoretic corrections: this is not even discussed in the text. line 10 and in the text: what is hyper-local? - In line 126, the subscript $m_k$ for the MoE does not make sense. The sum is over k, so why the subscript? - There is no composite likelihood in Eq (4) so the DGP model is not really an extension of the GP model. This is a shame as it would be interesting to compare both. - What happened with the notation $Y_{a,n,p}$ in Eq (4), there is no $p$ here? - Why is there a superscript in $X_^{(k)}$? why $X$ depends on k here? - line 88: replace n with N ### Significance ### Since the models are a combination of existing ideas and no new inference methods are proposed the significance of the theoretical/algorithmic side is low. The proposed method can have a high significance in the practical side but a better evaluation considering more datasets and more suitable baselines must be done.

[Author Response · NeurIPS 2019]

Table 1: Requested additional comparisons. SAT: Satellite, LAQN: Ground stations. Random seed of 0.

| Model | Data Sources | sRMSE ($\mu \pm \sigma$) | RMSE ($\mu \pm \sigma$) | NLPL ($\mu \pm \sigma$) |
|---|---|---|---|---|
| Single GP | LAQN only | $1.04 \pm 0.04$ | $23.02 \pm 11.26$ | $12.0 \pm 12.22$ |
| MR-GP | SAT only | $0.72 \pm 0.41$ | $14.87 \pm 9.34$ | $16.7 \pm 23.14$ |
| VBAgg-Normal | LAQN & SAT | $0.82 \pm 0.48$ | $16.24 \pm 9.15$ | $9.78 \pm 11.97$ |
| MR-GPRN w/o CL | LAQN & SAT | $0.69 \pm 0.43$ | $14.03 \pm 8.93$ | $9.24 \pm 14.35$ |
| MR-GPRN w/ CL | LAQN & SAT | $0.69 \pm 0.42$ | $14.45 \pm 9.09$ | $8.83 \pm 12.92$ |
| MR-DGP | LAQN & SAT | $\mathbf{0.39 \pm 0.13}$ | $\mathbf{8.65 \pm 4.93}$ | $\mathbf{4.54 \pm 4.12}$ |

We thank the reviewers for their time and detailed, constructive feedback. We are glad to see our application-motivated
methodological contributions and narrative were well-received. We denote with e.g. **R1.2.3** our response to Reviewer
1, Section 2, Paragraph 3. (**Joint**): As requested we are offering additional baselines (Table 1) that strengthen our
results and are discussed below. As suggested by **R3** we will move inducing point material to the appendix. The
additional page will allow us to improve clarity: we will expand on the MR-DGP model, we will add the additional
baselines, suggested by **R1** and **R4**, with further discussions of results and the uncertainty quantification benefits of the
CL corrections. We will also improve the description of the experiments and lighten the use of inline equations.

**R1.2.4**: We will improve the motivation for the composite likelihood (CL). The estimated CL ensures that the asymptotic
posterior $p(\mathbf{Y}|\mathbf{f}, \mathbf{X}, \theta)$ converges to the misspecified asymptotic MLE distribution [24, 30]. The CL cannot be set as a
free parameter because otherwise we would not obtain this theoretical guarantee. The MR-DGP learns dependencies
between the layers $p(\mathbf{f}_1|\mathbf{f}_2, \ldots)$ and hence between resolutions. **R1.2.5**: We have rerun all our experiments and
additional requested baselines on the real world example, see [**Joint**] above. **R1.2.7** The size of the std is due to
variability across the 42 sites in London, we will also offer site-standardized results (e.g. sRMSE) in the appendix. We
are happy to follow alternative standardizations if reviewers express a preference for the final version.

**R3.2.3**: We agree that the distinction between multi-fidelity and multi-resolution would be beneficial and we will offer
that. See [**Joint**] above to see how we will improve the explanation of MR-DGP (including the choice of kernel).
MR-DGP arises very naturally in real world examples and is able to successfully handle data from biased sensor
networks; as shown in our experiments the performance is significant. **R3.2.10-11**: We will reduce the number of inline
equations by moving the standard results into the appendix. **R3.2.13**: This is an interesting paper that simply takes the
formulation of [27] and applies it to the multi-task setting through the LCM formulation. The model proposed by the
authors is a special case of MR-GPRN where the latent GPs $\mathbf{W}$ are constant. We have also presented a very natural
and principled method to dealing with bias whereas they consider a very ad-hoc solution through data normalization.
**R3.5.1**: It is indeed natural that specific contributions will be more or less interesting to different readers. In this paper
we have tackled some of the underlying issues of previous approaches and offer the state-of-the-art.

**R4.1.1**: We respectfully disagree with "generalization of multi-task ... straightforward", in fact we have challenged
two very common assumptions by correcting (MR-GPRN) or accounting (MR-DGP) for dependent observations and
have provided a principled way to deal with biases and multi-resolution. Through these contributions we have shown
impressive results on a very complex problem. **R4.1.2 + R4.2.Originality**: Both are latent variable models and with
MR-DGP we use the latent structure to model the obs.dependency instead of correcting via CL. Further extensions
or special cases of this framework we leave for future work. **R4.2.3**: Indeed the composite weight comes out of the
expectation (See Eqn 14 in appendix). We do not model the cross-resolution dependencies via CL in the MR-GPRN
model, the weight corrects for posterior contraction due to loss of that dependency as done in similar settings [24]. We
will further demonstrate the UQ benefits of CL in the extra page through coverage and pred.densities. (**R4.2.4**) VBAgg
is the only published work that is a suitable baseline. We have shown that both handle the same types of multi-res
data in Sec. F of the appendix. As also suggested by **R3** we will merge this into the main text. We do not use [6] as
baseline because, despite the name, it is unable to handle multiple observation processes, see ($\ell : 170 - 173$). **R4.2.5**:
See [**R1.2.4**] above. **R4.2.6**: The dimension of the Hessian is $|\hat{\theta}| \times |\hat{\theta}|$ where $\theta$ are the hyper parameters. The size is
very small and is dominated. We will clarify this in the main text. (**R4.2.Clarity.3**): Thank you for pointing out the
typo in bolding MR-DGP, we will fix this. MAPE is an asymmetric loss that penalizes overestimation. As shown in Fig.
2 the prediction from MR-DGP is slightly over estimating whereas VBAgg-Normal is severely underestimating, hence
MAPE over penalizes MR-DGP. **R4.2.Clarity.4**: Different sensor networks are calibrated differently, hence comparing
raw values is not viable. The information theoretic corrections are from the composite weights in Sec. 3, we will clarify.
Hyper-local is higher resolution than typical LSOA area estimates. (**R4.2.Clarity.5-7**): Thank you for pointing out
these typos, we will fix them. The subscript $m_k$ is meant to be $m_a$ which represents the output for each layer. The $p$ is
used to denote the multiple tasks and because we have presented MR-DGP in the general case the ordering of tasks
between layers is a user-choice. We will improve clarity throughout the paper based on all reviewers' suggestions.

[Meta-Review · NeurIPS 2019]

This paper proposes a new GP-based method to handle real-world signals that are non-stationary, multi-task, multi-fidelity, and multi-resolution. The main proposals are two versions based on a shallow-GP and deep-GPs. The idea is to use a composite likelihood based approach to be able to handle such signals. All reviewers find the problem relevant and the solutions also useful. Experiments are done reasonably well, although there are some recommendations (eg from R3). There is a serious concern from R4 on the effectiveness of the composite likelihood. The paper is not written for non-experts, e.g., I consider myself an expert on GPs but not very familiar with GP-RN, and I am unable to follow the papers. It takes a few iterations to understand the details. This has also been pointed out by R1. The opinion of the paper has generally improved after the rebuttal and reviewers have increased their scores. The paper is still borderline but can be acceptable. I will encourage the authors to improve aspects of the paper to have a bigger impact, otherwise this paper might only be accessible to a narrow audience. I will give an accept to this paper for now, and hope that the authors will improve the paper taking the reviewers feedback into account.